# Ultrafast carbon monoxide photolysis and heme spin-crossover in myoglobin via nonadiabatic quantum dynamics

Konstantin Falahati [1], Hiroyuki Tamura[2], Irene Burghardt [1] & Miquel Huix-Rotllant [1,3]

Light absorption of myoglobin triggers diatomic ligand photolysis and a spin crossover transition of iron(II) that initiate protein conformational change. The photolysis and spin crossover reactions happen concurrently on a femtosecond timescale. The microscopic origin of these reactions remains controversial. Here, we apply quantum wavepacket dynamics to elucidate the ultrafast photochemical mechanism for a heme–carbon monoxide (heme–CO) complex. We observe coherent oscillations of the Fe–CO bond distance with a period of 42 fs and an amplitude of ~1 Å. These nuclear motions induce pronounced geometric reorganization, which makes the CO dissociation irreversible. The reaction is initially dominated by symmetry breaking vibrations inducing an electron transfer from porphyrin to iron. Subsequently, the wavepacket relaxes to the triplet manifold in ~75 fs and to the quintet manifold in ~430 fs. Our results highlight the central role of nuclear vibrations at the origin of the ultrafast photodynamics of organometallic complexes.

[1] Institute of Physical and Theoretical Chemistry, Goethe University Frankfurt, Frankfurt 60438, Germany. [2] Department of Chemical System Engineering, The University of Tokyo, Tokyo 113-8656, Japan. [3] Aix Marseille Univ, CNRS, ICR, Marseille, France. Correspondence and requests for materials should be addressed to I.B. (email: burghardt@chemie.uni-frankfurt.de) or to M.H.-R. (email: miquel.huixrotllant@univ-amu.fr)

Hemeproteins containing a porphyrin-iron complex (heme) play a major role in the storage and transport of diatomic molecules[1]. The diatomic ligand dissociation occurs concurrently to a spin-crossover (SCO) transition from low-spin (LS) to high-spin (HS) of the $Fe^{II}$ center[2]. The heme complex with carbon monoxide (CO) in myoglobin is one of the most studied hemeproteins of this kind[3–18]. Heme–CO is initially in a singlet LS state ($S = 0$), which transforms to a quintet HS state ($S = 2$) upon CO dissociation[10]. The unbound CO in the myoglobin cavity along with the motions of the remaining heme initiate a "protein quake" that opens a channel for the diatomic molecule to escape[3]. Over a century ago, Halden and Lorrain discovered that the CO dissociation can be initiated photochemically[19], by a mechanism which is still under debate. The photolysis can be initiated upon absorption to the lowest singlet porphyrin band ($^1Q$) or the second porphyrin band ($^1B$)[20]. Here, we focus on the photochemistry of the heme–CO complex upon excitation to the lowest Q-band of porphyrin.

The heme-CO photolysis is an ultrafast process. Recent pump-probe X-ray experiments of myoglobin, with an initial pump pulse exciting the $^1Q$ state, seem to agree on a two-step kinetic reaction: (i) a first step taking <50–70 fs, attributed to both CO photolysis and partial SCO, and (ii) a completion of the spin transition to the HS state in ~300–400 fs[5,6,20]. Despite the numerous studies on heme-CO photolysis, the kinetics and mechanism of dissociation are still under debate, notably regarding the ultrafast nature of the reaction, and the spin and character of the photolytic state. The rate of photolysis has never been experimentally reported, apart from the upper bound of 50–70 fs. As far as the photolytic state is concerned, the most widely accepted hypothesis is that dissociation occurs from a metal-to-ligand charge-transfer (MLCT) state[15–18,20]. Still, experiments and theory do not provide a unified picture to date. On the one hand, Franzen, Martin et al., based on time-resolved absorption and Raman experiments, described a "rapid spin state change that must precede photolysis"[20]. They consider that photolysis is occurring from a triplet metal→porphyrin ring transfer ($^3MLCT$ dissociation)[20,21]. In the model by Franzen et al., the ultrafast reaction is due to valence tautomerism. This mechanism implies a rapid interconversion of several quasi-degenerate electronic states involving d-transitions, some of which are dissociative for the CO bond. On the other hand, Head-Gordon et al., employing time-dependent density-functional theory, proposed a dissociation in a singlet metal → $\sigma^*_{CO}$ state ($^1MLCT$ dissociation)[15–18]. In the model of Head-Gordon et al., CO photolysis occurs in a Marcus-like process, in which the initial $^1Q$ population is transferred to the photolytic state after crossing a barrier of <0.2 eV. However, this hypothesis seems in contradiction with experimental observations of absence of fluorescence emission, an ultrafast reaction and a unity quantum yield for the heme-CO complex[22]. The theoretical method used provided an incomplete description of ultrafast heme-CO photolysis, since the coupled electron-nuclear motion was neglected. Recent theoretical studies of similar iron complexes indicate the fundamental role of nuclear motions in explaining the ultrafast nature of intersystem crossing (ISC)[23–25]. Such electron-nuclear strong coupling has been experimentally observed in time-resolved X-ray absorption in a Fe(II) tris(2,2'-bipyridine) complex[26,27] and also in the ferricyanide ion[28].

Here, we resolve the mechanism of ultrafast heme–CO photolysis by means of quantum wavepacket dynamics, which accounts for nuclear and electronic coherent motion[29]. We find that CO photolysis occurs in around 20 fs in the $^1MLCT$ band, prior to the spin transition. The ultrafast nature of the photolysis is due to strong vibronic couplings and a band of quasi-continuous states. Upon dissociation, the SCO process induces a

sequential transition of the remaining heme first into the triplet $^3MLCT$ and second to the HS quintet manifold $^5MLCT$.

## Results

**Heme–CO model and electronic structure.** The model taken as representative of the active center of the myoglobin protein consists of an iron(II) encapsulated in a porphyin ring, and axially ligated to a CO and an imidazole. Hereafter this model is referred to as heme–CO for the complex with carbon monoxide and simply heme for the remaining ligands. The LS state of the heme–CO complex has an electronic configuration $(a_2)^2(a_1)^2$ $(e_x^*, e_y^*)^0$ for the porphyrin and $(d_{xy})^2(d_{xz}, d_{yz})^4(d_{z^2})^0$ $(d_{x^2-y^2})^0$ for Fe. In its minimum energy structure, the iron and porphyrin are coplanar and the CO is bound to Fe in an upright position perpendicular to the plane. The Fe–C(O) bond is 1.80 Å, around 0.26 Å shorter than the Fe–N bonds with porphyrin or imidazole, which are 2.03 Å and 2.06 Å, respectively. Upon photolysis, the Fe changes into a quintet state, with electronic configuration $(d_{xy})^1(d_{xz}, d_{yz})^3(d_{z^2})^1$ $(d_{x^2-y^2})^1$. In the HS, the Fe–CO bond is unstable. The iron is displaced from the porphyrin plane, distorted to a square-pyramidal structure. The Fe–N bond length is 2.10 Å for the porphyrin nitrogens and 2.17 Å for imidazole.

The full electronic spectrum of heme–CO in the Franck-Condon region is shown in Table 1. This spectrum has been computed in the gas phase, at the CASSCF(10,9)/CASPT2/ANO-RCC-VDZP level of theory for the minimum energy structure of the $S_0$ ground state (the active space is shown in Supplementary Fig. 1). More details can be found on the effect on the spectrum of the basis set (Supplementary Table 1), the level of theory (Supplementary Table 2) and the model (Supplementary Tables 3–5). At this geometry, the HS state is found 0.7 eV above the LS state. The bright states are the $^1Q$ states, which correspond to a $\pi \to \pi^*$ transition localized on the porphyrin ring. The $^1Q_x$ and $^1Q_y$ states have an oscillator strength of $1.37 \cdot 10^{-2}$ and $9.13 \cdot 10^{-3}$ au for $Q_x$ and $Q_y$, respectively. In our model, these transitions are found at 2.73–2.79 eV, while

**Table 1 Electronic spectrum of the heme–CO model in the gas phase**

| State | Energy | Character | Sym. |
|---|---|---|---|
| $1^5MC$ | 0.70 | $(d_{yz}, d_{xy}) \to (d_{x^2-y^2}, d_{z^2})$ | E |
| $2^5MC$ | 0.74 | $(d_{xz}, d_{xy}) \to (d_{x^2-y^2}, d_{z^2})$ | E |
| $1^3MC$ | 0.89 | $d_{xy} \to d_{x^2-y^2}$ | $B_2$ |
| $3^5MC$ | 1.40 | $(d_{xz}, d_{yz}) \to (d_{x^2-y^2}, d_{z^2})$ | |
| $2^3MC$ | 1.43 | $d_{xz} \to d_{z^2}$ | E |
| $3^3MC$ | 1.51 | $d_{yz} \to d_{z^2}$ | E |
| $4^3MC$ | 1.62 | $d_{xy} \to d_{z^2}$ | $A_2$ |
| $5^3MC$ | 1.86 | $d_{yz} \to d_{x^2-y^2}$ | E |
| $6^3MC$ | 1.90 | $d_{xz} \to d_{x^2-y^2}$ | E |
| $1^1MC$ | 1.90 | $d_{xy} \to d_{x^2-y^2}$ | $B_2$ |
| $1^3MLCT$ | 2.15 | $(a_1, d_{yz}, d_{xy}) \to (e, d_{x^2-y^2}, d_{z^2})$ | |
| $2^1MC$ | 2.35 | $d_{yz} \to d_{z^2}$ | E |
| $3^1MC$ | 2.36 | $d_{xy} \to d_{z^2}$ | E |
| $1^3Q_1$ | 2.43 | $a_2 \to e$ | E |
| $^5MLCT$ | 2.48 | $(a_2, d_{yz}, d_{xy}) \to (e, d_{x^2-y^2}, d_{z^2})$ | |
| $2^3Q_2$ | 2.49 | $a_2 \to e$ | E |
| $1^3MLCT$ | 2.56 | $(a_1, d_{yz}, d_{xy}) \to (e, d_{x^2-y^2}, d_{z^2})$ | |
| $^1Q_x$ | 2.73 | $a_2 \to e/a_1 \to e$ | E |
| $^1Q_y$ | 2.79 | $a_2 \to e/a_1 \to e$ | E |
| $^1MLCT$ | 2.83 | $(a_2, d_{yz}) \to (e, d_{z^2})$ | |

Electronic states up to the bright $^1Q_{x,y}$ state are shown. The excitation energies (in eV) are computed at the CASSCF(10,9)/CASPT2/ANO-RCC-VDZP level of theory at the $S_0$ minimum energy structure obtained at the B3LYP/LANL2DZ level of theory. In addition, the state nature, the character of the dominant transitions and the symmetry label is shown

experimentally these bands are found at 2.14–2.30 eV[30]. These difference are not unexpected, since on the one hand we do not take into account the electrostatic environment of the protein, and on the other hand the statistical average of several protein conformations. Quantum Mechanics/Molecular Mechanics (QM/MM) calculations decrease the overall spectrum by around 0.84 eV for the lowest part of the spectrum. More importantly, the energetic order and the gap between the states in our model as compared with QM/MM is essentially conserved (Supplementary Table 6). This is essential to guarantee a realistic dynamical treatment.

The lowest valence states (up to ca. 2 eV) correspond to dark doubly-degenerate metal-centered (MC) states, involving transitions from the occupied $d_{xy}$, $d_{xz}$ and $d_{yz}$ orbitals to the unoccupied $d_{z^2}$ and $d_{x^2-y^2}$ orbitals. The crystal field splitting of the iron orbitals corresponds to a square pyramidal complex. The MLCT band is formed by states of mainly MLCT character, although some MC and LMCT character is also observed (Supplementary Table 7 for a detailed analysis of the state character). These bands of states have strong multi-configurational character, and were not correctly represented in previous single-reference studies based on density-functional theory[15,18].

**Quantum dynamics.** Quantum wavepacket dynamics has been performed using a vibronic model containing the main vibrational coordinates (heme doming, symmetry-breaking, rotational and dissociative modes, see Supplementary Fig. 2 for a plot of the vibrations and Methods section for further details.) The evolution of diabatic state populations during the first 500 fs is shown in Fig. 1a. The results have been obtained by sampling 10 independent quantum dynamics simulations with different initial conditions. The initial conditions are obtained by projecting the heme–CO geometries extracted from molecular dynamics snapshots of a myoglobin protein onto our vibronic model Hamiltonian (Supplementary Table 8.) In the Figure, only the states which are mainly populated are shown (Supplementary Fig. 3 for the dynamics of all states.) Following the dynamics of each spin manifold, a sequential transfer is found to occur, first from singlet to triplet (S → T) and then from triplet to quintet (T → Q). The time constant for S → T transfer is estimated as 76 ± 15 fs, while the T → Q decay time is 429 ± 70 fs, in perfect agreement with the experimental rates of Cammarata et al. and Franzen et al.[5,6,20] (for the timescales for each initial condition, see Supplementary Table 9 and Supplementary Note 1.) In the singlet manifold, we observe an abrupt decay of the $^1Q$ population, which is mainly transferred to the singlet $^1$MLCT manifold. The rate for this transfer is estimated to 26 ± 7 fs. The overall reaction is thus sequential $^1Q \rightarrow {}^1$MLCT $\rightarrow {}^3$MLCT $\rightarrow {}^5$MLCT (see Fig. 1b). The initial step is a complete transfer from the $^1Q$ state to the $^1$MLCT state, resulting in a negligible $^1Q$ population after ~100 fs. During the remaining transfer steps the three spin manifolds coexist. This is because the three manifolds are close in energy and strongly mixed through spin-orbit coupling. However, a clearly dominant spin state prevails during different intervals: singlet for t < 75 fs, triplet for 75 < t < 425 fs and quintet at later times.

In Fig. 1c, we show the evolution of the Fe–C(O) and the Fe out-of-plane distances for the first 500 fs (see Supplementary Figs. 4–6 for the corresponding evolution per set of initial conditions). Clearly, we observe damped large amplitude stretching motions for the Fe–C distance, which converges to a quasi-stationary value of 2.2 Å. These motions are coherent, with a period of 42 fs. The Fourier transform of this signal shows a frequency of 800 cm$^{-1}$, much faster than the Fe–C(O) stretching frequency of 488 cm$^{-1}$. In 21 fs, the Fe–CO distance

oscillates between the equilibrium distance (1.7–1.8 Å) and 2.5 Å. These are the typical distances at which the ground-state dissociation occurs[7]. After the first oscillation, the wavepacket is essentially in the MLCT bands, which are repulsive for CO such that the CO does not recombine and undergoes continued oscillations. Relaxation of the structure damps these oscillations until an equilibrium value of 2.2 Å is reached. Both the equilibrium value and the coherent frequency might be disrupted by the presence of the protein, an effect not included in the present model. This would introduce a faster decoherence of the Fe and CO interactions than what we observe in our model. As for the evolution of the Fe out-of-plane motion, we observe oscillations of ±0.2 Å around the initial position, in good agreement with recent time-resolved X-ray crystallography data[4]. As expected, the iron center reacts to the elongation of the Fe–CO distance by following the CO. Very rapidly, it oscillates back to the porphyrin plane, where part of the energy is dissipated to histidine. In the second and subsequent oscillations, the oscillations of Fe and CO are out of phase, indicating that the bond is photolyzed. A large dispersion of the Fe–C distances, which is appearing after 0.2 ps, is due to the delocalization of the wavepacket over the $^{1,3,5}$MLCT bands, indicating an unbound CO molecule. This dispersion is less marked for the Fe out-of-plane distance due to the more localized nature of the Fe atom, bounded strongly to the heme and the proximal histidine.

Experimental evidence shows that the CO is photolyzed within 70 fs[5,6,20]. In order to determine whether dissociation occurs from the singlet or the triplet state, we have considered a reduced model with only the singlet manifold, in which we eliminate the coupling parameters to the triplet and quintet states (Supplementary Figs. 7–9). In this case, an ultrafast transfer from Q → $^1$MLCT is still observed. The Fe–C(O) distance again exhibits oscillations with an amplitude of 0.7 Å. The fact that the amplitude is smaller than in the full model can be ascribed to the absence spin-orbit couplings which introduce strong mixing of the $^1$MLCT surfaces with $^3$MLCT states, that extend the region of oscillations. From these results, we can infer that the transfer to the $^1$MLCT manifold is sufficient to dissociate the CO, although the presence of $^3$MLCT favors further the photolysis. The dissociation is thus happening between 0.5 and 1.5 periods of the Fe–C(O) oscillation (20–60 fs), when the wavepacket is mainly in the $^1$MLCT state.

After photolysis, the SCO mechanism brings the system sequentially to a HS state. After ~75 fs, the $^3$MLCT band is clearly dominant, although a residual $^3$MC participation is also observed (Supplementary Fig. 1). From the triplet manifold, a population transfer to the HS states is slowly building up, until it becomes dominant at around 400 fs. At this time, the wavepacket is distributed across the $^5$MLCT and $^5$MC excited states, staying trapped in these states. This is consistent with the slow time constant of ~3 ps observed in myoglobin, which has been attributed to a vibrational cooling due to the protein[20,31]. Such excited state trapping has been observed recently in similar iron complexes[27]. This trapping can be understood in terms of the potential energy surfaces depicted in Fig. 2. Upon excitation to the $^1Q$ band, a sequential ultrafast transfer, first to $^1$MLCT and then to $^3$MLCT occurs. These transfers are energetically favored and therefore extremely fast. In the $^3$MLCT manifold, the wavepacket relaxes in a distribution of $^3$MLCT and $^3$MC states. The $^3$MC state can undergo direct transfer to the $^5$MLCT state, which is, however, energetically less favorable (and thus slower) than the singlet → triplet MLCT. The transition from the lowest $^5$MLCT to the $^5$MC state is energetically unfavoured. On the other hand, the $^3$MC states are energetically separated from the rest of the states, such that a transfer to the $^5$MC manifold becomes inefficient.

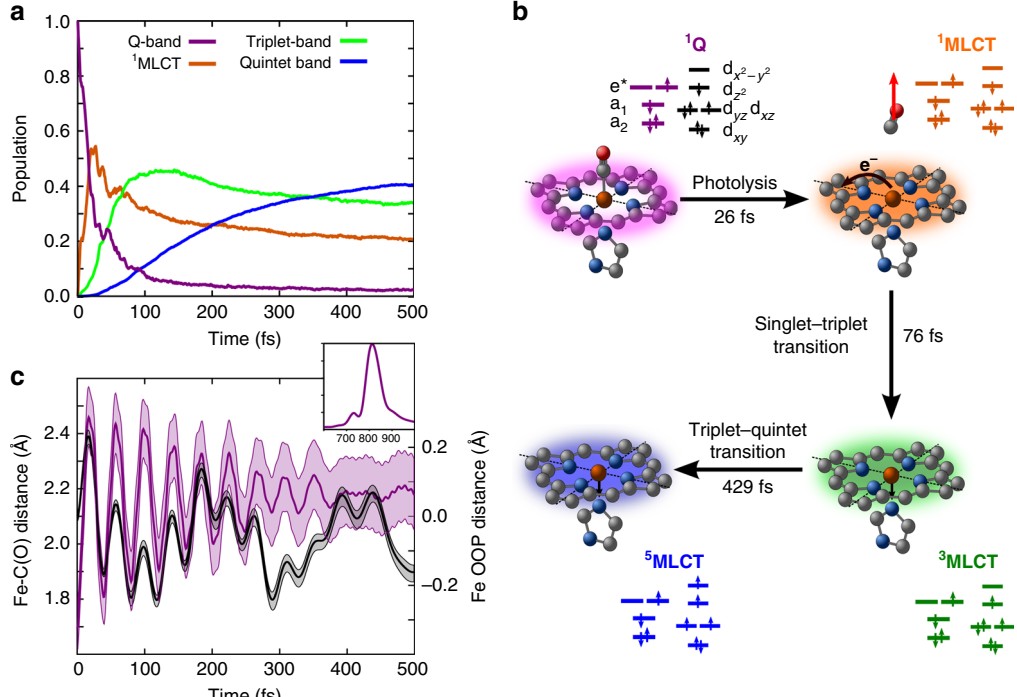

**Fig. 1** Photodynamics of photolysis and spin-crossover. Quantum photodynamics of heme–CO complex during the first 0.5 ps, with initial conditions averaged over 10 molecular dynamics snapshots. **a** Evolution of diabatic populations for states $^1$Q (magenta), $^1$MLCT (orange), triplet band (green) and quintet band (blue). The $^1$Q population rapidly decays giving rise to $^1$MLCT population dominating by 75 fs, at which point the triplet population increases. The quintet population builds up more slowly, and evolves into the dominant state at around 350 fs. **b** Schematic representation of the reaction mechanism and interpretation in terms of time constants. Upon initial excitation to the Q-band, the metal-to-ligand charge transfer (MLCT) state is populated in ~25 fs. In a second step, the system relaxes to the triplet (~75 fs) and to the lowest quintet state (~430 fs). Black arrows indicate the direction of the electron transfer and the main nuclear motions. **c** Evolution of the Fe–C(O) distance (magenta, left axis) and the Fe out-of-plane distance (black, right axis). Large amplitude motions are observed with a period of oscillation of 40 fs. The amplitude of oscillation is initially 0.9 Å and converges towards a value of 2.2 Å. At this distance, the CO is essentially photolyzed. The standard deviation of these geometric values is shown as a shaded area. In the inset, the Fourier transform of the Fe–C(O) oscillations is shown (in cm$^{-1}$)

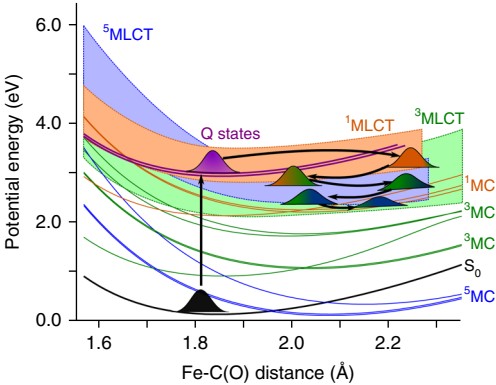

**Fig. 2** Potential energy surface along a dissociative mode. Plot of the potential energy surfaces along the Fe–C(O) distance. The LS state S$_0$, and the $^1$Q states are shown in black and magenta, respectively. The singlet manifold is shown in orange, the triplet manifold in green and the quintet manifold in blue. The metal-centered (MC) states are shown explicitly, whereas the metal-to-ligand charge transfer (MLCT) states are depicted as a quasi-continuous band. Further, a schematic evolution of the wavepacket dynamics is shown. After absorption to the Q-band, the wavepacket undergoes large-amplitude oscillations in the Fe-CO coordinate on the $^1$MLCT state. Then, it cascades down acquiring more $^3$MLCT and $^5$MLCT character as the spin crossover transitions occur. Finally, the wavepacket disperses, and the Fe–C(O) distance oscillations decrease to a value of 2.2 Å

**Vibronic effects**. The initial structure is close to a C$_{4v}$ symmetric geometry, in which the CO is in upright position and the porphyrin ring is close to the square planar conformation. The origin of the heme–CO bond is the typical $\sigma$-donation $\pi$-back-donation mechanism. A partial electron transfer from an occupied $\sigma$ orbital of CO to an empty d$_{z^2}$ orbital of Fe occurs simultaneously to a partial back transfer of electron density from the occupied d$_{yz}$ and d$_{xz}$ of Fe to the empty $\pi^*_{y,CO}$ and $\pi^*_{x,CO}$ orbital, respectively. At this geometry, the $\pi$ orbitals of porphyrin do not overlap with the d-orbitals.

Different types of vibrations can weaken the heme–CO bond, thus liberating the diatomic molecule. In Fig. 3, a schematic representation of the most representative vibrations is shown (for a detailed analysis of all vibrations, see Supplementary Table 10). The most important vibration is the dissociative coordinate (Fig. 3a), which corresponds to a symmetric Fe–CO stretching. This is the main reaction coordinate for CO photolysis. The Fe–CO stretching decreases the overlap between the iron and CO orbitals, thus decreasing both the strength of $\sigma$-donation and the $\pi$-back-donation. Once the CO is released, the remaining complex is then stabilized in the quintet state. This brings the Fe center out-of-plane, stabilized by $\pi$-interactions with the porphyrin antibonding orbitals.

Activation of the dissociative vibration is fundamental for releasing the CO from the heme. This vibration is of a$_1$ symmetry, as it is a vibration along the principal symmetry axis. However, the initial photon is absorbed exclusively in the porphyrin moiety.

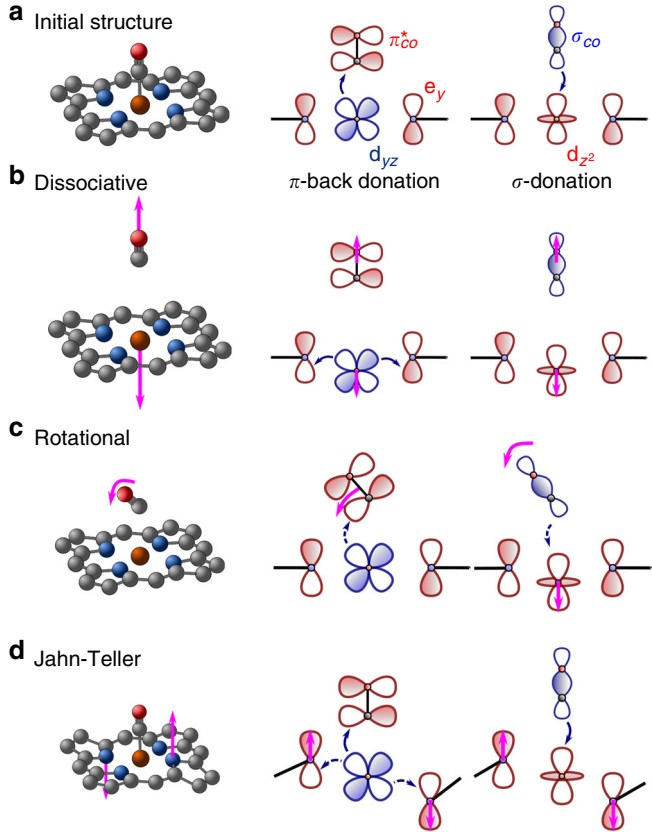

**Fig. 3** Vibrational motions and effect on the Fe-CO interaction. Representation of the nuclear motions (left) and the effect on the Fe-CO orbital interaction (right). The $\sigma_{CO}$ and $d_{xz}/d_{yz}$ orbitals (in blue) are occupied, while $e_{x/y}$, $\pi^*_{x/y,CO}$ and $d_{z^2}$ orbitals (in red) are unoccupied. Blue arrows indicate the direction of the partial electron transfer, due to the overlap between an occupied and an unoccupied orbital (solid and dashed arrows indicate strong and weak overlap). Magenta arrows indicate the vibrational motion of the specified atoms. **a** In the initial structure, the Fe-CO bond is strongest due to partial electron transfer $d_{xz}/d_{yz} \rightarrow \pi^*_x/\pi^*_y$ ($\pi$-back donation) and $d_{z^2} \leftarrow \sigma_{CO}$ ($\sigma$-donation). **b** Dissociative vibration, which corresponds to a stretching of the Fe-CO bond. This structure destabilizes the Fe-CO bond and stabilizes by a partial electron transfer $d_{xz}/d_{yz} \rightarrow e^*_x/e^*_y$ when the Fe atom is out of plane. **c** Rotational vibrations, which correspond to the rotation of the CO moiety. This vibration decreases the overlap between Fe and CO orbitals. **d** Symmetry-breaking vibration, in which the porphyrin is distorted inducing Jahn-Teller and pseudo Jahn-Teller vibronic couplings in the excited state. This vibration has no effect on the Fe-CO bond, but activates the coupling between the porphyrin and the Fe densities

The porphyrin optical Q-state is E-symmetric, thus localized in the perpendicular plane with respect to the principal axis. This implies that the dissociative vibration is not activated in the quasi-$C_{4v}$ initial geometry. The dissociation of CO thus requires an initial energy transfer from the porphyrin plane to the Fe center, which can only be possible by populating symmetry breaking vibrations. In the excited states, strong symmetry breaking of Jahn-Teller (JT) and pseudo Jahn-Teller (pJT) type occurs[32,33]. The JT effect is the sudden increase of inter-state couplings between an E-degenerate state along symmetry breaking vibrational coordinates. These JT couplings induce a double-well minimum in the potential energy surfaces at elongated coordinates, while the energy is maximum at the initially symmetric structure. The pJT effect is analogous to the JT effect but applies to non-degenerate states, usually requiring more than a single vibration and more than one electronic state. Both mechanisms are characterized by strong vibronic couplings. In the particular case of the heme–CO complex, the E-symmetry of the Q-band can be broken by $b_1$ and $b_2$ vibrations, in the so-called $E \otimes b$ JT mechanism[32]. A representative vibration of this type is depicted in Fig. 3d. Such a vibration is not directly coupled to the Fe–CO bond, but is fundamental to induce the interstate coupling between the Q-band and the MLCT band. The relaxation energy induced by such vibrations can be as large as 0.5 eV, which rapidly leads the wavepacket far from the Franck-Condon region. Large amplitude motions have been experimentally observed in similar iron complexes[26].

The rotation of CO (see Fig. 3c) has been discussed at length in the literature, generating a controversy on whether it has an effect on the photochemistry[8,9,16,34]. The rotation of CO slightly weakens the Fe–CO bond due to a less efficient overlap between the densities of each moiety. Furthermore, this rotation induces a Renner-Teller type of coupling (rovibronic coupling), in which e-symmetry bending modes of CO and $a_1$-type vibrations couple E and $A_1$ states, in what is known as the $(E \oplus A_1) \otimes (e \oplus a_1)$ effect. This mechanism can also couple the Q-band (E symmetry) and the MLCT band ($A_1$ symmetry) to the CO bending (e symmetry) and the dissociative iron-CO stretching ($a_1$ symmetry). The population evolution of the singlet manifold excluding the JT modes— and thus the rotational modes that are responsible of the Q-MLCT coupling—shows a similar population and Fe–C kinetics. This indicates that any symmetry breaking coordinate, related to protein fluctuations, rotations or symmetry breaking Jahn-Teller effects, will accelerate the coupling to the MLCT band.

The porphyrin doming ($a_1$ vibration of the porphyrin ring, see modes 8,12 in Supplementary Fig. 3) has been pointed out as an important motion for the photodynamics[5,6,35]. The doming motion is out-of-plane, and has a slow period of 371 fs. It is of the same symmetry as the dissociative mode, to which it is strongly coupled. The potential energy surface cuts along the doming mode indicate a weak reorganization energy and a small vibronic coupling for all potential energy surfaces. Therefore, even though this motion is necessary to stabilize the high-spin state, it plays a passive role and does not significantly affect the short-time dynamics of photolysis (Supplementary Figs. 10–12). Therefore, we conclude that the doming motion is in fact a consequence of the CO dissociation, that is strongly coupled to the Fe–CO dissociation coordinate, and is activated during the second kinetic step of 280 fs[5,6,20].

**Spin crossover.** The photochemical spin crossover mechanism is an important reaction in many organometallic complexes, in which ligand chromophores allow a usually fast low-to-high spin transition of the complex. Recent theoretical and experimental studies for similar iron complexes have shown that ligands play a fundamental role in modulating the time constants and the electronic pathway during the photoreaction[4–6]. The spin cross-over mechanism is mediated by the relativistic spin-orbit coupling (SOC). Although this is a purely electronic transition, the vibrational effects on the SOC seem fundamental to explain the ultrafast nature of spin crossover in metallic complexes. In the particular case of heme–CO, the main SOC strengths are about 200–300 cm$^{-1}$, which are of similar size as the JT vibronic couplings. Therefore, internal conversion and intersystem crossing are competing mechanisms in the dynamical relaxation mechanism.

In the heme–CO complex, the initially populated Q-band of porphyrin exhibits a negligible SOC with any other state. In

general, SOC is strong when metal density is present in the electronic state. Therefore, the wavepacket is initially dominated by the symmetry breaking vibronic couplings in the singlet manifold, leading to the transition $^1Q \rightarrow {}^1MLCT$ and $^1Q \rightarrow {}^1MC$. Both the $^1MLCT$ and $^1MC$ have strong spin-orbit coupling strength with $^3MLCT$ and $^3MC$ states, respectively. In general, the SOC is strong when the selection rules $\Delta S = 1$ and $\Delta L = 1$ are obeyed. The latter condition implies that one of the d orbitals changes orientation between the triplet and the singlet state. For example, a strong coupling occurs between the singlet state of character $d_{xy} \rightarrow d_{x^2-y^2}$ with the triplet state of character $d_{xz} \rightarrow d_{x^2-y^2}$. We observe that singlet-triplet transitions occur at a faster rate in the MLCT band than within the MC states (Supplementary Fig. 4). At around 50 fs, the $^3MLCT$ population becomes dominant, at the expense of the singlet manifold populations. This is because there are a larger number of crossing points between MLCT states of different multiplicity than between MC states (see Fig. 2 and Supplementary Fig. 4). This results in a dominant population of $^3MLCT$ and $^5MLCT$ states during the spin-crossover transitions. The $^3Q$ band triplets, which were pointed out as playing a role in the photophysics of heme–CO, contribute negligibly to the dynamics[20]. The $^1MC$ state exhibits a residual population after 100 fs, following prevalent transfer to the $^3MC$ state, with a population which is two times smaller than the $^3MLCT$ population. Finally, the $^3MLCT \rightarrow {}^5MLCT$ transfer builds up population in the quintet band. A $^3MC \rightarrow {}^5MC$ transfer, however, is not observed, due to the fact that the $^3MC$ and $^5MC$ states do not cross in an accessible region. Our results show rather that the wavepacket remains trapped in the $^5MLCT$ band. The trapping occurs in regions of the energy surface where relaxation to lower states (in this case MC states) is slow due to large gaps and small vibronic couplings, whereas a band of electronic states acts as an energy dissipation force, thus localizing the wavepacket in an excited state. Such observation for heme–CO is in line with the valence tautomerism model of Franzen, Martin et al.[20] In summary, the existence of quasi-continuous MLCT bands degenerate to the $^1Q$ state and strong vibronic couplings are at the origin of the ultrafast photolysis and spin-crossover in the heme–CO complex.

## Discussion

The mechanism of heme–CO photolysis and spin crossover has been elucidated by means of high-dimensional quantum dynamical calculations. The results indicate that the vibronic mechanism can be summarized in terms of a sequential $^1Q \rightarrow {}^1MLCT/MC \rightarrow {}^3MLCT \rightarrow {}^5MLCT$ transfer. During the first step, which is completed within 20–60 fs, the CO is dissociated, prior to the spin-crossover mechanism. The dissociation occurs in the singlet MLCT excited states. For these states, the $\sigma - \pi$ bond between Fe and CO bond is weakened. Subsequently, the wavepacket relaxes stepwise to the quintet band under the effect of spin-orbit coupling. We assign the three experimental rates observed for the heme–CO complex in myoglobin, recently reported in Ref. [20] and Refs. [5,6] as follows: (i) the 50–70 fs time constant corresponds to the singlet-triplet spin crossover, in which the population is mainly in the $^3MLCT$ excited state; (ii) the 300–400 fs time constant corresponds to the triplet-quintet spin crossover, in which the population is mainly in the $^5MLCT$ excited state; (iii) the 2.4 ps time constant we tentatively assign to the relaxation of the $^5MLCT$ excited state to the final HS state of the heme.

## Methods

**Electronic structure**. Unconstrained structure optimization and normal-mode calculations of $S_0$ in gas phase have been performed with density-functional theory (DFT) using the B3LYP exchange-correlation functional and the LAN2DZ basis

set[36–39]. These computations have been performed with Gaussian09[40]. Excited states have been computed with CASSCF(10,9)/CASPT2[41,42]. The active orbitals are the Fe(II) 3d-orbitals and the Gouterman's porphyrin $\pi$-orbitals (see Supplementary Fig. 2 and Supplementary Note 2). State-averaged calculations have been performed for 35 singlet states, 50 triplet states, and 30 quintet states. In all calculations, we employed relativistic Douglas-Kroll-Hess (DKH) Hamiltonian with the ANO-RCC-VDZ basis set[43–45]. Larger basis sets do not significantly affect the energetics (Supplementary Table 2). Multi-state CASPT2 has been performed with frozen core and 116 frozen virtual orbitals. Similarly to coupled-cluster[46], deleting virtual orbitals accelerates computations and keeps the essential physics (Supplementary Table 2). In particular, the spin crossover transition is well represented and the DFT minima are also CASPT2 minima. Spin-orbit couplings have been obtained by expanding the DKH Hamiltonian in the basis of 20 singlets, 20 triplets and 20 quintets[45]. All multi-configuration results have been obtained with Molcas[47].

**Vibronic Hamiltonian**. The model Hamiltonian for representing the dissociation and the spin crossover is of the form:

$$\hat{H}(\mathbf{q}) = \sum_{S=0}^{2} \hat{H}_S(\mathbf{q}) + \sum_{S,S'=1}^{2} \hat{V}_{SS'}^{SO}, \quad (1)$$

in which $S = 0, 1, 2$ is the total spin quantum number, $\mathbf{q}$ is the mass-frequency-weighted vibrational coordinate vector, $\hat{V}^{SO}$ is the spin-orbit coupling. The vibronic Hamiltonian, represented in the basis of ground-state normal modes, is given by

$$\hat{H}_S(\mathbf{q}) = \hat{T}_S(\mathbf{q}) + \hat{V}_S(\mathbf{q}), \quad (2)$$

containing the harmonic kinetic energy and potential operator. The potential is given by

$$\hat{V}_S(\mathbf{q}) = \sum_{m_s=-|S|}^{|S|} \sum_{I,J=1}^{20} \left[\delta_{IJ}E_I^S + V_{IJ}^S(\mathbf{q})\right] \left|{}^S\Psi_I^{m_s}\right\rangle\left\langle{}^S\Psi_J^{m_s}\right| + \text{c.c.}, \quad (3)$$

in which $m_s$ is the azimuthal spin and $\left|{}^S\Psi_I^{m_s}\right\rangle$ is the state wavefunction. The $E_I^S$ energies are diabatic at the $S_0$ minimum geometry; at the quasi-$C_{4v}$ geometry these can be considered quasi-diabatic. The $V_{IJ}^S(\mathbf{q})$ potential is derived from one-dimensional potential energy surface cuts which were diabatized by fitting to the following functional form,

$$V_{II}^S(\mathbf{q}) = \sum_{i=1}^{5} \frac{1}{i!}\frac{\partial^i E_I^S(\mathbf{q})}{\partial \mathbf{q}^i}\Big|_{\mathbf{q}=0}\mathbf{q}^i$$
$$+ \boldsymbol{\alpha}_I^S e^{\boldsymbol{\beta}_I^S(\mathbf{q}-\gamma_I^S)} + \mathbf{D}_I^S\left(1 - e^{\varepsilon_I^S(\mathbf{q}-\mathbf{q}_I^S)}\right)^2, \quad (4)$$

$$V_{IJ}^S(\mathbf{q}) = \mathbf{v}_{IJ}^S(\mathbf{q})\mathbf{q}. \quad (5)$$

Here, the inter-state couplings $V_{IJ}^S(\mathbf{q})$ are approximated within a linear vibronic model. The set of parameters are determined by a least square fitting

$$f(\boldsymbol{\gamma}^S) = \frac{1}{N}\sum_{I\sigma}\left(E_I^{mod,S}(\{\boldsymbol{\gamma}_I^S\},\mathbf{q}_\sigma) - E_I^{PT2,S}(\mathbf{q}_\sigma)\right)^2, \quad (6)$$

where $\boldsymbol{\gamma}^S = \left\{\frac{\partial^i E_I^S}{\partial \mathbf{q}^i}, \boldsymbol{\alpha}_I^S, \boldsymbol{\beta}_I^S, \boldsymbol{\gamma}_I^S, \varepsilon_I^S, \mathbf{q}_I^S, \mathbf{v}_{IJ}^S\right\}$, $\mathbf{q}_\sigma$ is the discretized value of a normal mode coordinate and $E_I^{mod,S}(\mathbf{q}_\sigma)$ and $E_I^{PT2,S}(\mathbf{q}_\sigma)$ are the spin-free adiabatic energies from the model Hamiltonian and the CASPT2 calculation, respectively. The latter are obtained by one-dimensional potential energy surface cuts along each ground-state normal mode direction. The passive normal modes have been kept frozen along the cuts.

The fitting of quasi-continuous bands requires an accurate initial guess. We employ the following protocol: (i) quasi-diabatic surfaces are obtained by maximum overlap of CI coefficients; (ii) pre-fitting is done on the quasi-diabatic surfaces by employing first and second order Taylor expansion or the exponential function, (iii) the adiabatic PT2 surfaces are fitted with the full diagonal Hamiltonian, and (iv) the full Hamiltonian is fitted until the total error is <0.10 eV.

**Vibrational modes**. Vibrational coordinates have been chosen by performing one-dimensional potential energy cuts and selecting the modes with largest reorganization energy. There are a total of 15 explicit vibrational modes in the model: 1 symmetric and 1 asymmetric Fe-CO stretchings, 2 doming modes, 1 CO stretching, 1 in-plane, and 1 out-of-plane porphyrin vibrations, and 4 doubly degenerate modes (Jahn-Teller modes of porphyrin out-of-plane, a porphyrin in-plane mode and a pure CO bending.) These modes have been selected in order to reproduce the symmetry breaking that allows the Q → MLCT ultrafast transition, the dissociation and the relaxation of the quintet heme. A schematic plot of these vibrations can be found in Supplementary Fig. 3. Duschinsky rotation effects and

anharmonic couplings between vibrational modes have been neglected in our model; in particular, the latter are negligibly small compared to the vibronic couplings (see Supplementary Table 11 and Supplementary Note 3).

**Quantum dynamics**. The Multi-Layer Multi-Configuration Time-Dependent Hartree (ML-MCTDH) method[48,49] (Heidelberg package, version 8.5.5) was used to perform wavepacket propagation up to 1 picosecond for a vibronic coupling model comprising 179 electronic states of singlet, triplet and quintet multiplicities, and 15 vibrational modes exhibiting strong vibronic coupling. A four-layer representation of the wavefunction was employed, where 8 single-particle functions (spf) were chosen per layer, with a primitive harmonic-oscillator discrete variable representation (DVR) comprising up to 80 DVR points (see Supplementary Note 4 and Supplementary Fig. 13). Initial conditions were constructed in accordance with the relative oscillator strengths of the singlet ligand and metal-centered states at the Franck-Condon geometry.

## Data availability
The authors declare that the data supplementary the findings of this study are available within the paper and its supplementary information files. Parameters for the model Hamiltonian and potential energy surfaces along each vibrational mode are available from the corresponding authors upon request.

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

## Acknowledgements

MHR acknowledges financial support from Agence Nationale de la Recherche (grant ANR-16-CE29-0008, BIOMAGNET) and the Alexander von Humboldt-Foundation. K. F. and I.B. gratefully acknowledge funding by the Deutsche Forschungsgemeinschaft via RTG 1986 "Complex Scenarios of Light Control". I.B. thanks Prof. Gerhard Hummer (MPI Biophysics, Frankfurt) for making MD snapshots available for the present study. All authors thank Prof. Nicolas Ferré, Dr. Padmabati Mondal, Dr. Anthony Kermagoret and the reviewers for useful comments. This work was granted access to the HPC resources of Aix-Marseille Université financed by the project Equip@Meso (ANR-10-EQPX-29-01) of the program "Investissements d'Avenir" supervised by the Agence Nationale de la Recherche.

## Author contributions

M.H.R. designed the research in collaboration with I.B. M.H.R., I.B., and H.T. developed the vibronic Hamiltonian. M.H.R. and K.F. performed the electronic structure calculations. M.H.R., K.F., and I.B. performed the quantum dynamics. M.H.R. wrote the paper. All authors contributed to the discussions and gave comments on the manuscript.

## Additional information

**Competing interests:** The authors declare no competing interests.

