## [Peer Review File · Nature Communications]

Reviewers' comments:

Reviewer #1 (Remarks to the Author):

Marco Cammarata (CNRS)

author of the femtosecond XAS experiment cited by the authors

In this well written paper the authors performed state of the art quantum chemical calculations to obtain new insights on a classical photophysical system: Myoglobin complexed with carbon monoxide. Over the decades this system has been used as testbed to try to understand theoretically and to prove experimentally as much as possible of the photophysics.

In this regard this manuscript is a very nice contribution. In fact despite decades of research by different research groups the mechanism of photodissociation, the electronic and structural dynamics that accompanying it are still under debate in part due to extreme speed at which it happens that has prevented - so far - definitive experimental answers.

What I like about this paper is that it puts forward hypothesis that can be verified (or disproved) experimentally in the years to come. For this reason I think that it deserves publication in Nature Communications provided that few points are clarified/discussed/improved.

I will divide my comments in three groups: 'important', 'quite important' and 'minor' points. Within each group sorting has no particular meaning.

IMPORTANT COMMENTS (I*)

=====

I1: I think that one of the weakest points of the manuscript is the comparison with the experimental data, the authors insist so much on.

The way experimentalists try to summarize the experimental results varies but in the cited cases (time resolved Raman and my own X-ray absorption) the data are analyzed in terms of sum of exponentials (convoluted with the instrument response function). In this sense a reported Y fs cannot be usually interpreted as 'finished by Y fs' [as the authors sometime seem to imply] but rather in the exponential sense (complete after $\sim 3 \times Y$). How the authors obtain the time scales from their work is not very transparent and the manuscript would benefit from a clearer description.

I would not mind seeing discrepancy between 'expected from theory' and 'estimated from experiment' values but I don't like too much the current 'confusion' in which the experimental data seem to support the authors findings more than they probably are.

This is evident for example in the 70 fs comparison. The authors write (second column of second page) 'The triplet manifold becomes mainly populated after 70 fs ... This is in close agreement with the initial decay time obtained in time resolved ...'.

Similarly a bit further down the authors state 'At 280 fs, the quintet population is dominant, consistent with the second decay time estimated as 300 fs by both Franzen and Cammarata'.

Looking at the authors data ['Quintet band' of figure 1b] it seems that the characteristic Quintet formation is close to $1 - \exp(-t/100)$. In my own XAS experiment I observed a second exponential of the kind $1 - \exp(-t/400)$.

Similarly Franzen work [and actually the previous work by the same group, J. W. Petrich, Biochemistry, 1988, 27 (11), pp 4049–4060] cannot be used to compare the value precisely as the authors simply state that is too fast for their time resolution.

To clarify I would analyze the author results as done for experiments. This would allow a more direct comparison between experiment and theoretical findings.

This would be particularly useful for the Fe-C (and the Fe-out-of-plane, see below) distance that are a direct observable of X-ray techniques like femtosecond crystallography.

I2: what does the fig1b mean ? at the longest shown time there is not unique state but rather a combination of MLCT, triplet and quintet. How does this evolve for longer times ? what a 'non pure state' means ? that every protein is an ill-defined spin state ? or that in an ensemble we could find different molecules in different spin states ? The authors should clarify.

I3: The Fe-C distance plot (1C) is very interesting. Can the authors explain the step-like response at ~15 fs ? in other words it seems that the acceleration is very small until 15 fs and suddenly increases afterwards. I don't seem to see such a clear step in the fig 1A. A comment would be very beneficial. Possibly even having fig1B and 1C with shared time axis (from 0 to 1ps) with inset for the short time ?

I4: sometimes (for example Barends et al Science 2015) the iron-out-of-plane motion is calculated. A plot of this quantity would be very useful.

I5: related to the point above: recently (in the same ref 11) there have been QM/MM simulations that have calculated the Fe displacement. how do the authors results compare with those simulations ? Is the Fe displacement similar ? If not why ? I think that a sentence in the manuscript is essential.

I6: have the authors tried to run calculations without triplet and quintet states ? They should be faster to run (less states) but they could be very informative about the dissociation. How would the Fe-C distance change ? The authors allude to such calculations (top second column, second page) but I could not find the relevant information in the Supp. Mat.

I7: If I am not mistaken the results have been obtained from a single calculation that assumes certain initial positions. At room temperature the ensemble of molecules probed in experiments span a vast conformational space. How such heterogeneity could impact the dynamics is not clear to me. Without knowing how much computational effort is what I am asking I was wondering if it could be possible to repeat the calculation for few more initial states. This would give an estimate of one of the contribution to the uncertainty to the timescales the authors report.

I8: still referring to figure 1C, we have recently published indications of the sensibility of XAS to the de-localization of the wavepacket (Lemke et al, Nat Comm 2017). In other words, certain x-ray photon energies have resulted particularly sensitive to the width of the distribution of the ensemble. Displaying a moving standard deviation could help quantifying the expected effect in MbCO again with one day the hope to be able to measure it (via XAS or diffraction). The same should be done for the Fe-out-of-plane.

I9: the authors correctly stress the importance of the symmetry of the vibrational modes. I think this point can be made a lot clearer with a figure displaying the most important states.

I10: I find figure 3 potentially useful (even very useful) but also unclear. The authors should try to improve the caption at the very least (what are the green arrows ? and the red crosses ?).

I11: nowhere in the manuscript the role of the protein is actually discussed. A comment about what the protein matrix can change in the dynamics would be useful.

QUITE IMPORTANT (QI*)

=====

QI1: the authors state that the rotational modes of the CO do not significantly influence the

dynamics as the vibronic coupling are smaller than the JT ones. Have an author a proof of that ? or a reference ?

Q12: somewhat related to the point above: the authors 'demonstrate' that the porphyrin ring (mode 8) is not a key player. The 'demonstration' lies on cuts of the potential energy surface (figures SI5-SI11) that are extremely hard to read. What are the different lines ? What should I learn from them ?

Q13: the description of the singlet spectrum is somewhat hard to follow (first column, second page). A plot with labels for the most important transitions (color coded somehow ?) would be very beneficial. A double x-axis eV/wavelength would also be nice. A comparison with an experimental spectrum would also be beneficial.

Q14: [related to I2] the authors state that they have performed calculations for 1 ps but never shown anything longer than 300 fs. I would add those data somewhere. If possible in the main text [depending also on how informative they are].

Q15: the authors refer to different modes (8,35,36), although reported in the SI I think the paper would benefit from a clearer image of those modes in the main text. The mode drawing table (table SI VIII) is too small. While ok for most of the modes, the most important ones deserve a better image, possibly in the main text.

MINOR COMMENTS (M*)

=====

M1: The authors write (first column, first page): "The unbound CO in the myoglobin cavity initiates the 'protein quake' ", I would add that the out of plane motion of iron is also responsible for the launching of the quake.

M2: fig1 caption: "The green line indicates a moving average ...". How long is the window ? which shape [triangular, gaussian, etc] ?

M3: capitalization is wrong in many references, for example co → CO

M4: ref 30 publication year is 2017

M5: ref 34 first authors is C. Rovira

Reviewer #2 (Remarks to the Author):

This report describes the application of quantum wave packet theory to the excited state dynamics of heme ligated to carbon dioxide. The focus is to understand the photolysis of CO from heme. This is a complex process since an excitation in a pi-pi* transition of the porphyrin ring ends up causing the Fe-C bond to break. It has long been known that the d-orbitals are involved in the excited state due to their mixing with the pi and pi* states of the heme, which ultimately leads to population of metal-to-ligand charge transfer (MLCT) state. The study examined which vibrational modes are coupled to the dissociative process. The conclusion that the axially symmetric Fe-C stretch is coupled is to be expected. However, the vibronic modes are less well understood. The conclusion that a Renner-Teller-type Fe-C-O degenerate bending mode only has weak coupling is a relevant conclusion since certain experimental studies have indicated a rotational trajectory (e.g. those of Anfinrud and co-workers). However, is the rotational trapping of the CO in myoglobin then solely a function of the protein structure? This is difficult to know based solely on the wave packet study, which only informs us of the earliest events (i.e. first 300 fs). The coupling of the other out-

of-plane vibronic ring mode would be difficult to anticipate from experimental data. However, the fundamental conclusion of the study that the photolytic event takes place in the singlet manifold was already anticipated by Franzen et al. based on the experimental data from both time-resolved absorption and resonance Raman spectroscopy.

Therefore, the conclusions of the paper are not surprising. And in terms of the mechanistic conclusions there is little that is new. On the other hand, this study represents a very nice high level study of the problem that brings a level of theory to bear on the problem that addresses the mechanism with a new level of detail. Because of the level of detail it is clearly worthy of publication. The question then becomes whether this study brings a sufficiently new insight that it warrants publication in Nature Communications. While I feel that the authors have done a good job of presenting some rather complex calculations in a format that is readable and relatively easy to understand, the manuscript does not bring a substantially novel insight to the problem. The detailed understanding of this problem would be appropriate to a journal such as Journal of Physical Chemistry or Biophysical Journal, which this publication should be readily accepted because of the nice application of high level theory to a problem of interest to the heme and biophysics communities.

Reviewer #1 (Remarks to the Author):

In this well written paper the authors performed state of the art quantum chemical calculations to obtain new insights on a classical photophysical system: Myoglobin complexed with carbon monoxide. Over the decades this system has been used as tested to try to understand theoretically and to prove experimentally as much as possible of the photophysics. In this regard this manuscript is a very nice contribution. In fact despite decades of research by different research groups the mechanism of photodissociation, the electronic and structural dynamics that accompanying it are still under debate in part due to extreme speed at which it happens that has prevented - so far - definitive experimental answers. What I like about this paper is that it puts forward hypothesis that can be verified (or disproved) experimentally in the years to come. For this reason I think that it deserves publication in Nature Communications provided that few points are clarified/discussed/improved.

I will divide my comments in three groups: 'important', 'quite important' and 'minor' points. Within each group sorting has not particular meaning.

IMPORTANT COMMENTS (I)*

=====

*II: I think that one of the weakest point of the manuscript is the comparison with the experimental data, the authors insist so much on. The way experimentalists try to summarize the experimental results varies but in the cited cases (time resolved Raman and my own X-ray absorption) the data are analyzed in terms of sum of exponentials (convoluted with the instrument response function). In this sense a reported Y fs cannot be usually interpreted as 'finished by Y fs' [as the authors sometime seem to imply] but rather in the exponential sense (complete after $\sim 3*Y$). How the authors obtain the time scales from their work is not very transparent and the manuscript would benefit from a clearer description.*

I would not mind seeing discrepancy between 'expected from theory' and 'estimated from experiment' values but I don't like too much the current 'confusion' in which the experimental data seem to support the authors findings more than they probably are.

This is evident for example in the 70 fs comparison. The authors write (second column of second page) 'The triplet manifold becomes mainly populated after 70 fs ... This is in close agreement with the initial decay time obtained in time resolved ...'. Similarly a bit further down the authors state 'At 280 fs, the quintet population is dominant, consistent with the second decay time estimated as 300 fs by both Franzen and Cammarata'. Looking at the authors data ['Quintet band' of figure 1b] it seems that the characteristic Quintet formation is close to $1-\exp(-t/100)$. In my own XAS experiment I observed a second exponential of the kind $1-\exp(-t/400)$. Similarly Franzen work [and actually the previous work by the same group, J. W. Petrich, Biochemistry, 1988, 27 (11), pp 4049–4060] cannot be used to compare the value precisely as the authors simply state that is too fast for their time resolution.

To clarify I would analyze the author results as done for experiments. This would allow a more direct comparison between experiment and theoretical findings. This would be particularly useful for the Fe-C (and the Fe-out-of-plane, see below) distance that are a direct observable of X-ray techniques like femtosecond crystallography.

Response: In response to the reviewer, we have now fitted the population evolution with exponentials and extracted the kinetic rates in a more transparent way. We used $\exp(-t/k)$ for the initial state, and $1-\exp(-t/k)$ for the final state. We explain precisely how these rates are calculated in Section III.F (page 27) of the Supporting Information (SI).

Since we performed a statistics of quantum dynamics simulations over 10 initial conditions extracted from molecular dynamics simulations (see page 15, Sec. III.B of the SI), we can even give a standard

deviation of the extracted kinetic rates. In Table IX, we summarize these kinetic rates obtained from our quantum dynamics simulation, and we added a discussion of the mean kinetic rates in the main text.

We think that with these newly calculated rates, the connection between our simulations and experimentally measurable rates has been clarified. In the previous manuscript, we performed a single quantum dynamics simulation starting from an initial condition structure that is highly symmetric (C_{4v}). For this simulation, singlet-triplet rate (τ_{S-T}) is obtained as 60.6 fs, which compared reasonably to the experimental first kinetic value (50-70 fs). The triplet-quintet rate (τ_{T-Q}) of 252.8 fs, was underestimated with respect to the second kinetic rate (300-400 fs).

Now, with the statistics over 10 initial conditions, we obtain the following mean rates:

$$\tau_{S-T} = 76.3 \pm 15.9 \text{ fs} \quad \tau_{T-Q} = 429.2 \pm 70.1 \text{ fs}$$

The good agreement with experiment allows us to directly relate our rates to the first and second kinetic experimental rates. This gives a certitude that our model qualitatively and semi-quantitatively reproduces the experimental short time dynamics.

I2: what does the fig1b mean ? at the longest shown time there is not unique state but rather a combination of MLCT, triplet and quintet. How does this evolve for longer times ? what a 'non pure state' means ? that every protein is an ill-defined spin state ? or that in an ensemble we could find different molecules in different spin states ? The authors should clarify.

Response: This was perhaps not precisely explained in the previous manuscript. We observe decoherence of the wavepacket, which causes a population distribution over the different spin manifolds. Indeed, the interpretation is rather that in the protein we can find an ensemble molecules in different spin states. In this respect, the dominant population will dominate the experimental signal, although smaller signals of the other spin manifolds cannot be excluded. The “non pure state” in this context was a misleading term, and we have eliminated it. We have added the following discussion in the text:

“The only complete transfer is the 1Q to the 1MLCT state, which ends up in a negligible population after a 100 fs. For the rest of transfers the three spin manifolds coexist. This is because the three manifolds are close in energy and strongly mixed through spin-orbit coupling. However, a clear dominant spin is appearing: singlet for $t < 76$ fs, triplet for $426 \text{ fs} > t > 76$ fs and quintet afterward.”

I3: The Fe-C distance plot (1C) is very interesting. Can the authors explain the step-like response at ~15 fs ? in other words it seems that the acceleration is very small until 15 fs and suddenly increases afterwards. I don't seem to see such a clear step in the fig 1A. A comment would be very beneficial. Possibly even having fig1B and 1C with shared time axis (from 0 to 1ps) with inset for the short time ?

Response: We have replaced this Figure, which now includes the statistics of the Fe-C distance. When investigating the origin of the step reported previously, we realized that this was due to a bug in our analysis tools. We have now resolved this and find much more plausible results, where the step no longer appears. The amplitude of the oscillations is slightly larger, due to the initial condition sampling

that we performed (see further I7). Now, a clear coherent oscillation of the Fe-C is appearing. By using discrete Fourier transforms, we extracted the frequency of oscillation to be 800 cm^{-1} . This effective oscillation frequency in the excited state is much larger than the frequency of stretching of the ground state, which we found at 488 cm^{-1} . The 800 cm^{-1} corresponds to a period of 42 fs. Based on the analysis of the Fe out-of-plane distance (see responses I4), we can estimate that the dissociation takes between 20-60 fs. We have added the following discussion in the manuscript:

“Experimentally, it is evidenced the CO is already photolyzed within 70 fs.^{5,6,18} In order to determine whether this dissociation occurs from the singlet or the triplet state, we have considered a reduced model with only the singlet manifold, in which we set to 0 the coupling parameters to triplet and quintet states (see Supporting Information). In this case, an ultrafast transfer from $^1\text{Q}\rightarrow^1\text{MLCT}$ is observed. However, the Fe-C(O) distance oscillates only around 0.1-0.2 Å. This indicates that the spin crossover is essential to the dissociation. The Fe-C(O) distance oscillations are due to the anharmonicities induced by the $^1\text{MLCT}$ - $^3\text{MLCT}$ spin-orbit coupling. From these analysis, we can conclude that the dissociation occurs in the triplet manifold. This is happening in 1.5 periods of Fe-C(O) oscillation (ca. 60 fs), when the wavepacket is majoritarily in the $^3\text{MLCT}$. However, we cannot exclude the possibility of the $^1\text{MLCT}$ dissociation. The initial oscillation of nearly 1 Å occurring in the first 20 fs would be enough to dissociate this bond, if the energy of CO would be dissipated by the protein environment. This effect is presently not included in our model.”

I4: sometimes (for example Barends et al Science 2015) the iron-out-of-plane motion is calculated. A plot of this quantity would be very useful.

Response: From our simulations, we can easily extracted the iron out-of-plane motion. Since we oriented the His-Fe-CO axis in the z-direction, we have simply calculated the iron-out-of-plane distance as the difference between the Fe position along the z-direction at a given time subtracted from its relaxed value at time 0. This is explained in the SI (see Sec. III.E, Figs. 12 and 13, pages 25-26), where we give the dynamic evolution of the Fe out-of-plane for each of the dynamics. We have added the mean value over 10 dynamics simulations in the new Figure 1c (see green line.) The value oscillates between $\pm 0.2\text{ Å}$, in perfect agreement with the molecular dynamics simulations of Barends et al. Science 2015. We observe that this value correlates to the Fe-CO distance oscillations. Indeed, in the first oscillation, when the CO is detached, the Fe centre follows up. In the second oscillation, they are out of phase. This indicates that already during the second oscillation the bond is dissociated. We have added the following discussion in the text:

“From the evolution of the Fe out-of-plane motion, we observe oscillations of $\pm 0.2\text{ Å}$ out of its initial position, in good agreement with the recent time-resolved X-ray crystallography.⁴ As expected, the iron centre to the elongation of the Fe-CO distance by following the CO. Very rapidly, it oscillates back to the porphyrin plane, where part of the energy is dissipated to histidine. In the second and subsequent oscillations, the oscillations of Fe and CO are out of phase, indicating that the bond is photolyzed.”

I5: related to the point above: recently (in the same ref 11) there have been QM/MM simulations that have calculated the Fe displacement. how do the authors results compare with those simulations ? Is the Fe displacement similar ? If not why ? I think that a sentence in the manuscript is essential.

Response: We obtained very similar displacements for the shortest time, in full agreement with X-ray crystallography and QM/MM calculations. This is to be expected, as the Fe out-of-plane distance is in fact well captured by our model and it is independent of the protein environment. For the corresponding modifications of the text, see response of point I4.

16: have the authors tried to run calculations without triplet and quintet states ? They should be faster to run (less states) but they could be very informative about the dissociation. How would the Fe-C distance change ? The authors allude to such calculations (top second column, second page) but I could not find the relevant information in the Supp. Mat.

Response: We have added these calculations in the SI (see Sec. III.C.1 and Fig. 4 in pages 15-16). Even though the evolution looks very similar on the shortest time, the signals saturate very rapidly with a main population in the ¹MLCT state in all cases. We also observe that with less symmetric initial conditions, the population of MC states is less important. We have further added the information on the Fe-C distance for all these dynamics in the SI (see Sec. III.D.1 and Fig. 8 in pages 22) along with the relevant information for the Fe out-of-plane (see Sec. III.F.1 and Fig. 12 in pages 25). Both follow a similar evolution as in the full model, although the amplitude of motion is smaller by 0.3 Å. This is related to the fact that we restrict the wavepacket motion over a band of 14 microstates in the singlet simulation, while in the full model the wavepacket moves on a manifold of ca. 150 microstates.

17: If I am not mistaken the results have been obtained from a single calculation that assumes certain initial positions. At room temperature the ensemble of molecules probed in experiments span a vast conformational space. How such heterogeneity could impact the dynamics is not clear to me. Without knowing how much computational effort is what I am asking I was wondering if it could be possible to repeat the calculation for few more initial states. This would give an estimate of one of the contribution to the uncertainty to the timescales the authors report.

Response: In this revised manuscript, we have performed a statistical average of 10 quantum dynamics simulations. The initial conditions have been obtained by extracting 10 snapshots from the molecular dynamics simulations performed in the article of Baerends et al. Science, 2015. These snapshots have been projected in the normal modes in which we have constructed our vibronic model Hamiltonian. This is explained now in the SI (see Sec. III.B, page 15). We have also added each population evolution and all the structural analysis for each individual set of initial conditions (see Figs. 4-8 and 10-14 of the SI). In the main text, we show the mean value over all dynamics realizations of the diabatic populations, the kinetic rates and the dynamic evolution of Fe-C and Fe out-of-plane distances.

18: still referring to figure 1C, we have recently published indications of the sensibility of XAS to the delocalization of the wavepacket (Lemke et al, Nat Comm 2017). In other words, certain x-ray photon energies have resulted particularly sensitive to the width of the distribution of the ensemble. Displaying a moving standard deviation could help quantifying the expected effect in MbCO again with one day the hope to be able to measure it (via XAS or diffraction). The same should be done for the Fe-out-of-plane.

Response: We have added this in the new Figure 1c of the revised manuscript. We observe that while the dispersion of the Fe out-of-plane remains very small all along the 0.5 ps simulation, the Fe-C distance get sparse after 200 fs. We have added the following comment in the text:

“The large dispersion of the Fe-C distances, which is appearing after 0.2 ps, which appears due to the

delocalization of the wavepacket over the ^{1,3,5}MLCT bands. This delocalization is less important for the Fe out-of-plane distance.”

I9: the authors correctly stress the importance of the symmetry of the vibrational modes. I think this point can be made a lot clearer with a figure displaying the most important states.

Response: We have clarified and added more detailed information of the vibrational modes in Figure 3 of the Supporting Information. Their effect on the potential energy surfaces is explained also in the SI, Figures 15-21 (see response QI5 and QI2).

I10: I find figure 3 potentially useful (even very useful) but also unclear. The authors should try to improve the caption at the very least (what are the green arrows ? and the red crosses ?).

Response: We have remade this figure. We have taken out the green arrows and red crosses, which apparently were not quite clear to the reader. These were supposed to indicate the direction of electron transfer, due to a stronger/weaker overlap between occupied and virtual orbitals. In the new version of the manuscript, we have colored occupied orbitals as blue, and virtual orbitals as red. The direction of electron transfer is indicated by blue arrows (equivalent to the previous green arrows) and we have eliminated the crosses. We added also labels for each of the orbitals. We changed the caption to the following:

“FIG. 3. Vibrational motions and effect on the Fe-CO interaction. Representation of the nuclear motions (left) on the Fe-CO orbital interaction (right). The σ_{CO} and d_{xz}/d_{yz} orbitals (in blue) are occupied, while $e_{x/y}$, π^*_{CO} and d_{z^2} orbitals (in red) are unoccupied. Blue arrows indicate the direction of the partial electron transfer, due to the overlap between an occupied and an unoccupied orbital (solid and dashed arrows indicate strong and weak overlap). Magenta arrow indicates the vibrational motion of the specified atoms. (a) In the initial structure, the Fe#CO bond is strongest due to partial electron transfer $d_{xz}/d_{yz} \rightarrow \pi^*_{xy}$ (π -back donation) and $d_{z^2} \leftarrow \sigma_{\text{CO}}$ (σ -donation). (b) Dissociative vibration, which corresponds to a stretching of the Fe-CO bond. This structure destabilizes the Fe-CO bond and stabilizes by a partial electron transfer $d_{xz}/d_{yz} \rightarrow e_{x/y}$ when Fe is out of plane. (c) Rotational vibrations, which corresponds to the rotation of the CO moiety. This vibration decreases the overlap between Fe and CO orbitals. (d) Symmetry-breaking vibration, in which the porphyrin gets distorted inducing Jahn-Teller and pseud Jahn-Teller vibronic couplings in the excited state. This vibration has no effect on the Fe-CO bond, but it switches on the coupling between the porphyrin and the Fe densities.”

I11: nowhere in the manuscript the role of the protein is actually discussed. A comment about what the protein matrix can change in the dynamics would be useful.

Response: The effect of the protein environment at the chosen level (MS-CASPT2) is beyond the present computer capabilities. However, given the good comparison of the kinetic rates with experiments and QM/MM calculations for the Fe out-of-plane, we believe that the short time dynamics is well described by the heme-CO system. In order to confirm the quality of our gas phase results, we have performed a preliminary MS-CASPT2 calculation using QM/MM environment of the protein. Unfortunately, we were able to obtain the lowest part of the spectrum. We have included these results in the SI (see Sec. II.C, page 9). Indeed, although the absolute energies are very different (~0.8 eV on

average), the relative energies between the states are kept (~ 0.1 eV difference on average). While the absolute energetic position is not affecting the dynamics (in fact, the model Hamiltonian has been arbitrarily shifted so that the ground-state corresponds to the 0.0 eV energy), the relative energetics is directly affecting the wavepacket motion, as this will change the distribution of kinetic and potential energies accumulated in the wavepacket. More importantly, the QM/MM results show that the order of the MC states in our reduced model are the same as in the protein. We have added the following sentences in the manuscript:

“QM/MM calculations decrease the overall spectrum by around 0.84 eV for the lowest part of the spectrum. More importantly, the energetic order and the gap between the states in our model and with QM/MM is essentially conserved (see Supporting Information). This is essential to guarantee a realistic dynamical treatment.”

QUITE IMPORTANT (QI)*

=====

Q11: the authors state that the rotational modes of the CO do not significantly influence the dynamics as the vibronic coupling are smaller than the JT ones. Have an author a proof of that ? or a reference ?

Response: We made this statement based on a simple inspection of the average value of the JT couplings, which was larger than the rotational modes. In order to have a better proof of the effect of each normal mode on the dynamics, we have performed a dynamics of “reduced models” in which we have taken out certain classes of the model. This is shown in the Supporting Information in Fig. 4 for the population evolution, Fig. 8 for the evolution of the Fe-C distance, and Fig. 12 for the Fe OOP distance). These have to be compared with the full model with all vibrations as shown in the Supporting Information graphs entitled “Dynamics 1” (Fig. 3 for the population evolution, Fig. 7 for the Fe-C distance evolution and Fig. 11 for the Fe OOP distance evolution).

As can be seen, the population evolution and the geometric changes are almost the same when we exclude the JT modes and when we exclude the rotational modes. This implies that any symmetry breaking mode (either rotation or JT) accelerates the transfer from Q \rightarrow MLCT transition. In order to show this, we have also done a dynamics without any symmetry breaking mode (excluding modes number JT modes 16,17,21,22 and rotational modes 35,36).

We added the following sentence in the manuscript:

“The population evolution of the singlet manifold excluding the JT modes (and thus the rotational modes are responsible of the Q-MLCT coupling) shows a similar population and Fe-C kinetics (see Figs. 5 and 9 in the Supporting Information.) This indicates that any symmetry breaking coordinate (protein fluctuations, rotations or symmetry breaking Jahn-Teller) will accelerate the coupling to the MLCT band.”

Q12: somewhat related to the point above: the authors 'demonstrate' that the porfiring ring (mode 8) is not a key player. The 'demonstration' lies on cuts of the potential energy surface (figures SI5-SI11) that are extremely hard to read. What are the different lines ? What should I learn from them ?

Response: These plots show the adiabatic potential energy surfaces where the wavepacket can move. We have included them to show which modes are more anharmonic in the excited states. A stronger anharmonicity implies a large reorganization energy towards that mode direction, which largely dominates the wavepacket motion. We have added the following sentence in the supporting information:

“In this section, we plot the adiabatic potential energy surfaces of the ground and excited states for the singlet, triplet and quintet manifolds. The nature of the states at the 0 coordinate are detailed in Sec. II.G. Due to the symmetry of the normal modes, the state couplings can be activated. This can be observed as strong anharmonicities, which implies a strong displacement of the particular mode with the corresponding energy lowering (reorganization energy.)”

Q13: the description of the singlet spectrum is somewhat hard to follow (first column, second page). A plot with labels for the most important transitions (color coded somehow ?) would be very beneficial. A double x-axis eV/wavelength would also be nice. A comparison with an experimental spectrum would also be beneficial.

Response: We have added Table 1 in the main text, where we give the main transitions of the lowest part of the spectrum. A comparison with experimental absorption spectrum is at this point less interesting, since our model does not explicitly take into account the polarization due to the protein environment, and so we are 0.7 eV blue shifted. However, we mentioned what is the effect of the protein environment in the text and in the supporting information (see response I5).

Q14: [related to I2] the authors state that they have performed calculations for 1 ps but never shown anything longer than 300 fs. I would add those data somewhere. If possible in the main text [depending also on how informative they are].

Response: We have extended the Figures in the main text from 0.3 ps to 0.5 ps. Even though we carried out simulations for a duration of 1 ps, we believe that the results could be misleading to some extent. First, because the convergence of the numerical integrator for the wavepacket becomes more difficult. Second, because the effect of the protein becomes essential in the dynamics (see Baerends et al., Science, 2015). The purpose of the article is to describe the shortest time dynamics, and with our model we can correctly reproduce the time frame corresponding to the first two kinetic rates. In the future, we are planning to include the protein effects in the dynamics and the electronic structure to interpret the third kinetic rate. However, this is presently beyond our computational capacities.

Q15: the authors refer to different modes (8,35,36), although reported in the SI I think the paper would benefit from a clearer image of those modes in the main text. The mode drawing table (table SI VIII) is too small. While ok for most of the modes, the most important ones deserve a better image, possibly in the main text.

Response: We have added a schematic representation of the modes included in the vibronic model Hamiltonian in the Supporting Information. Since we are including 15 modes, and some modes involve very delocalized motions, we feel it is inappropriate to put it directly in the main text. Rather, we preferred to show in the main text classes of vibrations and their effect on the potential energy surfaces (see Fig. 3 and responses I9 and I10). We have added the Fig. 3 (page 15) in the Supporting information, and referenced it on the main text:

“c. *Vibrational coordinates*: Vibrational coordinates have been chosen by performing potential energy cuts and selecting the modes with largest reorganization energy. There are a total of 15 explicit vibrational modes in the model: 1 symmetric and 1 asymmetric Fe-CO stretchings, 2 doming modes, 1 CO stretching, 1 in-plane and 1 out-of-plane porphyrin vibrations, and 4 doubly degenerate modes (Jahn-Teller modes of porphyrin out-of-plane, a porphyrin in-plane mode and a pure CO bendings.) These modes have been selected in order to reproduce the symmetry breaking that allows the Q \rightarrow MLCT ultrafast transition, the dissociation and the relaxation of the quintet heme. The schematic plot of these vibrations can be found in Sec. III of the SI.”

MINOR COMMENTS (M)*

=====

M1: The authors write (first column, first page): "The unbound CO in the myoglobin cavity initiates the 'protein quake' ", I would add that the out of plane motion of iron is also responsible for the launching of the quake.

Response: Indeed, this is correct. We have added the following sentence in the revised manuscript:

“The unbound CO in the myoglobin cavity along with the motions of the remaining heme initiate the 'protein quake' [...].”

M2: fig1 caption: "The green line indicates a moving average ...". How long is the window ? which shape [triangular, gaussian, etc] ?

Response: This figure has been redone in the revised manuscript. The green line has been eliminated from the discussion, which we believe induced some confusion.

M3: capitalization is wrong in many references, for example co \rightarrow CO

Response: This has been corrected in the revised manuscript. We have revised all references in the manuscript.

M4: ref 30 publication year is 2017

Response: This has been corrected in the revised manuscript.

M5: ref 34 first authors is C. Rovira

Response: This is now Ref. 29 in the revised manuscript. It has been corrected.

Reviewer #2 (Remarks to the Author):

This report describes the application of quantum wave packet theory to the excited state dynamics of heme ligated to carbon dioxide. The focus is to understand the photolysis of CO from heme. This is a complex process since an excitation in a pi-pi transition of the porphyrin ring ends up causing the Fe-C bond to break. It has long been known that the d-orbitals are involved in the excited state due to their mixing with the pi and pi* states of the heme, which ultimately leads to population of metal-to-ligand charge transfer (MLCT) state. The study examined which vibrational modes are coupled to the dissociative process. The conclusion that the axially symmetric Fe-C stretch is coupled is to be expected.*

Response: The reviewer is right in saying that the Fe-C stretching was expected to be the main mode of dissociation. However, the focus of this study is in explaining the ultrafast origin of this reaction (<50-70 fs), which is far from obvious. Following simple symmetry rules, the polarization of the initial pi-pi* transition on porphyrin is localized in the xy plane, thus orthogonal to the dissociation coordinate (z-direction). The fastest modes are in-plane, which do not couple to the z-polarized states. The out-of-plane modes can couple xy-polarized states with z-polarized states, although they are very slow. We show that the Q→¹MLCT is occurring due to symmetry breaking modes of the Jahn-Teller type, which are out of plane and induce a large reorganization energy. These help not only to populate the ¹MLCT state but also to couple to the Fe-C stretching. To our knowledge, this is the first time that the Jahn-Teller effect is mentioned in the context of myoglobin dissociation.

Furthermore, we find that, the effective frequency of Fe-C stretching in the excited state is 800 cm⁻¹, much faster than the 488 cm⁻¹ of the ground state. In addition, we estimate the dissociation to occur within the first half period, thus in ~20 fs, when the wavepacket is still in the ground state. These values had never been reported before, neither experimentally nor theoretically.

However, the vibronic modes are less well understood. The conclusion that a Renner-Teller-type Fe-C-O degenerate bending mode only has weak coupling is a relevant conclusion since certain experimental studies have indicated a rotational trajectory (e.g. those of Anfinrud and co-workers). However, is the rotational trapping of the CO in myoglobin then solely a function of the protein structure? This is difficult to know based solely on the wave packet study, which only informs us of the earliest events (i.e. first 300 fs).

Response: One must not confuse the effect of rotational trapping in the ground-state with the effect of rotations in the excited state. This study shows that the Renner-Teller type couplings in the excited states are weaker (induce a smaller vibronic coupling) than the Jahn-Teller or the spin-orbit couplings, which are dominant. Therefore, the rotation of the CO is not a dominant mode for the wavepacket motion in the excited state. To clarify this point, we have added in the supporting information the dynamics of a reduced vibronic model without the Jahn-Teller modes and without the rotational modes, to compare the effect on the dissociation (see SI, Fig. 5, “Dynamics without modes 35,36”).

The coupling of the other out-of-plane vibronic ring mode would be difficult to anticipate from experimental data. However, the fundamental conclusion of the study that the photolytic event takes place in the singlet manifold was already anticipated by Franzen et al. based on the

experimental data from both time-resolved absorption and resonance Raman spectroscopy. Therefore, the conclusions of the paper are not surprising. And in terms of the mechanistic conclusions there is little that is new.

Response: The spin state at which the dissociation occurs is an open issue. Our results give the first convincing proof that this reaction is happening from the singlet manifold. It is not true that Franzen et al. anticipated that this reaction is happening on the singlet manifold. On the spin crossover, they say [Biophys. J., 80, 2372 (2001), page 2381]:

“The VT mechanism proposed here provides an explanation for the rapid spin state change that must precede photolysis.”

Thus, they clearly hypothesize that spin crossover occurs before the photolysis. Their article anticipated that the charge-transfer (CT) state is populated prior to dissociation, which we confirm in our article.

To our knowledge, the first hypothesis that this might happen from the singlet manifold was due to Head-Gordon et al [JACS, 124, 12070 (2002)]. However, we believe their conclusion was right for the wrong reason. Indeed, they chose a level of theory (TDDFT/B3LYP) which underestimates CT states. They find a low lying $d \rightarrow \sigma^*_{CO}$, which we do not find using the higher level of theory MS-CASPT2. Indeed, the same authors published a paper two years later on the failure of TDDFT for CT states [JACS, 126, 4007 (2004)].

On the other hand, this study represents a very nice high level study of the problem that brings a level of theory to bear on the problem that addresses the mechanism with a new level of detail. Because of the level of detail it is clearly worthy of publication. The question then becomes whether this study brings a sufficiently new insight that it warrants publication in Nature Communications. While I feel that the authors have done a good job of presenting some rather complex calculations in a format that is readable and relatively easy to understand, the manuscript does not bring a substantially novel insight to the problem. The detailed understanding of this problem would be appropriate to a journal such as Journal of Physical Chemistry or Biophysical Journal, which this publication should be readily accepted because of the nice application of high level theory to a problem of interest to the heme and biophysics communities.

Response: Even though our manuscript could be easily accepted in a physical chemistry or biophysics journal, we believe that our results can interest a broader community, and therefore we target multidisciplinary journals.

In our manuscript, we give the first direct proof that the photolysis is happening in the singlet manifold MLCT. Furthermore, we give the precise vibrational mechanism that explains this ultrafast photolysis, which is happening within 20 fs. These important results, reported for the first time, advance the knowledge in the microscopic mechanism of photolysis in myoglobin, and will hopefully will be proved experimentally when shorter-pulse X-ray lasers will be available.

Our simulation protocol, that can be applied to the photochemistry of any organometallic system. As

the reviewer acknowledged, we believe that the simulation protocol presented here will set the pace for future simulations of photochemistry of organometallic systems. This is now a flourishing field, ranging from radical photopolymerization, to spintronics, light-emitting diodes, photocatalysis, etc.

Reviewers' comments:

Reviewer #1 (Remarks to the Author):

The authors have addressed all my comments; the paper - that I found very interesting and well written already during the first submission - has much improved and I fully support publication in Nature Communications

Reviewer #2 (Remarks to the Author):

My original criticism of this paper is that the study merely provides detail for ideas that have already been confirmed both experimentally and computationally more than 16 years ago. In the revised version the authors state that the mechanism of photolysis is controversial. But, this is not correct. The authors have incorrectly suggested that there is a dichotomy between the models of Head-Gordon (2002) and Franzen et al. (2001). Franzen and Martin's model was very clear that the photolytic state had to be a singlet state. Franzen et al. based the nomenclature and spectroscopic information not only on the resonance Raman, but on transient absorption spectra obtained in Ref. 18 and the publications by Petrich and co-workers that preceded that work. Perhaps that confused the authors, but their suggestion that Franzen and Head-Gordon disagree is simply not correct.

The authors state that their study elucidates the importance of vibrational modes for photolysis. However, Figure 3 shows three modes (1) a Fe-CO stretch (2) a Fe-C-O bend and (3) a Jahn-Teller distortion. The authors state that the Jahn-Teller distortion is not important for photolysis, which is something that is obvious from the symmetry of the problem. One does not need a calculation at the CASSCF(10,9)/CASPT2/ANO-RCC-VDZP level in order to establish which mode is most important here. The point it is not clear what new information is being presented in this manuscript. Indeed, the wave packet study is a nice contribution that reminds us of an interesting problem, but the justification for publication in Nature is lacking.

Reviewer #3 (Remarks to the Author):

I have read the manuscript as well as the previous reports and rebuttals. To make it short:

I wish I would have done this work!

Why? It is not only time (there are many papers appearing on spin-crossover these days), but it also must be considered as providing a theoretical benchmark to the drosophila system for studying photo triggered protein dynamics.

Apparently, the response to reviewer #1 did improve the presentation substantially. W.r.t. reviewer #2, I obviously agree that this is excellent work, but I cannot share his/her view that the readership will come from a specialized audience only.

To wrap it up I recommend publication, provide that the authors have addressed the following points:

- literature:

There is a large body of work by Chris Meier on the quantum dynamics of this molecule & I wonder why this has been ignored. Further, concerning spin-crossover and the role of vibrations it might be worth citing the recent photoelectron study on Fe(CN)₆ by Aziz and coworkers in PCCP 19,

14248 (2017)

- technical:

I did not fully understand the model Hamiltonian. First, Eqs 4-5 suggest that anharmonic mode coupling has been taken into account in a rather unusual way, i.e. thinking of a Taylor expansion of the exponential. At the same time I doubt that this is actually what has been done since only diagonal cuts of the adiabatic potentials are represented.

If mode coupling has been neglected, please motivate!

How the neglected coordinates in the 1D cuts have been treated, i.e. frozen or relaxed? In any case, motivate the choice.

From the same equations I would gather that the dissociative coordinate is taken to correspond to the Fe-C(O) stretching normal mode. Practically, normal mode displacements are only meaningful in the vicinity of equilibrium structures. that means even small displacements of other atoms which are not part of the actual Fe-C(O) stretching, but exist because the Fe-C(O) motion is not fully decoupled, might cause some artificial behavior if the system is stretched too far. Is this perhaps the reason for the short time approximation or the limited range of distances in Fig. 2?

- presentation:

Fig. 1: I don't think two digit numbers are justified in panel b.

Fig 1. caption: "Period of oscillation is initially 0.9 A", What's meant here?

Fig. 1: the shaded area was not visible upon printing

line 140: Did the initial geometries break the symmetry? How about temperature? Is this supposed to mimic finite T as well?

line 149 and further on: The unit of a rate is 1/time, better call the numbers time constants or alike

line 171: Fe-C(O) stretching frequencies are apparently state dependent. What do we learn from this statement which is emphasized in several places?

line 401: "entropic trapping". I have difficulties to accept this characterization given the fact that a reduced dimensional model is considered at a very short time scale, i.e. there is little chance that possible micro states have been fully explored yet.

Response to reviewers (NCOMMS-17-25617B):

Reviewer #1 (Remarks to the Author):

The authors have addressed all my comments; the paper - that I found very interesting and well written already during the first submission - has much improved and I fully support publication in Nature Communications.

Response: We are very grateful for the reviewer's efforts in commenting our manuscript. All the remarks have been extremely valuable to improve the quality of our study. We added a note of appreciation in the acknowledgements.

Reviewer #2 (Remarks to the Author):

My original criticism of this paper is that the study merely provides detail for ideas that have already been confirmed both experimentally and computationally more than 16 years ago.

Response: Before addressing this critique, we would like to point out the different opinion that the reviewer expressed on the first version of our manuscript:

“this study represents a very nice high level study of the problem that brings a level of theory to bear on the problem that addresses the mechanism with a new level of detail. Because of the level of detail it is clearly worthy of publication.”

We cannot share the current opinion that our study “merely provides details” to previously confirmed ideas. The theoretical studies carried out 16 years ago did not confirm the experiments, since they neglected the main physical effect at play during photolysis: the electron-nuclear coupling. To quantitatively capture this effect, we have performed the most advanced simulation of heme-CO to date using quantum dynamics. We have given new information on the ultrafast nature of the photolysis, and we have clarified previous hypotheses from experiments and theory. Our conclusion is that photolysis occurs from a singlet state, that the photolytic state is of $d \rightarrow \pi^*$ character, and that vibronic couplings are responsible for the ultrafast nature of this reaction. For the first time, we have provided the rate of dissociation (~ 20 fs), the amplitude of elongation of the Fe-CO and the effective frequency of this motion in the excited state. Finally, we reproduced very accurately two experimental rates for the first time, which gives a strong confirmation of our model and hypotheses. We believe these are not minor details, but rather an important step towards the understanding of this complex photoreaction.

The seminal works conducted 16 years ago did not describe a unified picture of photolysis. Franzen et al. and Head-Gordon et al. both refer to intermediate charge-transfer states, but their interpretation in terms of multiplicity differs fundamentally (see below). The experiments of Franzen et al. led to a triplet dissociation model from a $d \rightarrow \pi^*$ and invoked a valence tautomerism as the origin of dissociation [Biophys. J., 80, 2372 (2001)]. The theoretical work of Head-Gordon et al. resulted on a singlet dissociation model from a $d \rightarrow \sigma_{CO}^*$ state in a Marcus-like process [J. Am. Chem. Soc., 124, 12070 (2002)].

In the revised version the authors state that the mechanism of photolysis is controversial. But, this is not correct. The authors have incorrectly suggested that there is a dichotomy between the models of Head-Gordon (2002) and Franzen et al. (2001). Franzen and Martin’s model was very clear that the photolytic state had to be a singlet state. Franzen et al. based the nomenclature and spectroscopic information not only on the resonance Raman, but on transient absorption spectra obtained in Ref. 18 and the publications by Petrich and co-workers that preceded that work. Perhaps that confused the authors, but their suggestion that Franzen and Head-Gordon disagree is simply not correct.

Response: As we already pointed out in our previous response, and as briefly summarized above, the (dis-)agreement between the models of Franzen et al. and Head-Gordon et al. seems to be a matter of confusion in the literature of heme-CO photolysis. To be more specific on this controversy:

Head-Gordon et al. concluded that photolysis happens in a charge-transfer (CT) type singlet state. In the introduction of their manuscript, they mention experimental evidences of singlet photolysis, citing the work of Franzen et al. We refer to the following highlighted part in the manuscript of Head-Gordon et al. *J. Am. Chem. Soc.*, 124, 12070 (2002):

The photodissociation of the CO-ligated species is a complicated process that starts with the excitation of the electronic singlet ground state into the Q_1 state, the energetically lowest $\pi-\pi^*$ transition which is well-known in porphyrins.⁵ Recent experimental findings indicate vary rapid decay into a second excited singlet state (E_1) on a time scale of about 50 fs, which is assumed to be the photodissociation step.⁴ The time scale of this decay step is too short to allow spin conversion. However, the excited unligated system decays then under spin conversion into its high-spin ground state on a longer time scale (picoseconds).⁴ Since we are interested

- 124, 12070.
 (4) Franzen, S.; Kiger, L.; Poyart, C.; Martin, J.-L. *Biophys. J.* **2001**, *80*, 2372
 (5) Rovira C.; Kunc, K.; Hutter, J.; Ballone, P.; Parinello, M. *J. Phys. Chem.*

Reprinted adapted with permission from *J. Am. Chem. Soc.*, 2002, 124 (41), pp 12070–12071. Copyright 2002 American Chemical Society.

This is a misinterpretation of the experimental work of Franzen et al., which does refer to ultrafast photolysis from a CT state, but says exactly the contrary regarding spin multiplicity. Textually from the article of Franzen, Martin, et al. [*Biophys. J.*, 80, 2372 (2001)]: “**rapid spin change that must precede photolysis**”. In other words, they state that photolysis occurs from a triplet state. The publication is online to confirm this [[https://doi.org/10.1016/S0006-3495\(01\)76207-8](https://doi.org/10.1016/S0006-3495(01)76207-8)]. Here we show a marked-up print-out of page 2381:

Fe(III), which has an electronic configuration of $[(d_{xz})^1(d_{xy})^1(d_{yz})^1(d_z)^1(d_{x^2-y^2})^1(a_{1u}, a_{2u})^4(e_g^*)^1]$. The VT mechanism proposed here provides an explanation for the rapid spin state change that must precede photolysis. The mechanism works for Hb^*O_2 as well. The charge transfer

Reprinted from *Biophysical Journal*, 80, Stefan Franzen, Laurent Kiger, Claude Poyart, Jean-Louis Martin, *Heme Photolysis Occurs by Ultrafast Excited State Metal-to-Ring Charge Transfer*, 2372–2385, Copyright 2001, with permission from Elsevier.

The causality between spin change and photolysis is mentioned in two other instances in the paper (left, page 2381 and right, page 2383):

Kunkely, 1976). The data presented here and other recent data support the hypothesis that Hb_1^* is a CT state. It follows that photolysis in Hb^*CO may occur due to formation of an intermediate spin CT state. For Hb^*CO and Hb^*NO , the $d_{\pi} \rightarrow a_{1u}, a_{2u}$ CT state results in destabilization of the

a photoinduced-CT process driven by the vacancy in the a_{1u}, a_{2u} HOMO. In Hb^*CO , the entropy change associated with an increase in spin multiplicity and population of low frequency modes drives a process that leads to spin conversion and destabilization of the bound CO ligand. We advance the hypothesis that VT is the mechanism for diatomic ligand photolysis in heme, as opposed to a previously proposed mechanism that involves a build-up of electron den-

Reprinted from *Biophysical Journal*, 80, Stefan Franzen, Laurent Kiger, Claude Poyart, Jean-Louis Martin, *Heme Photolysis Occurs by Ultrafast Excited State Metal-to-Ring Charge Transfer*, 2372–2385, Copyright 2001, with permission from Elsevier.

Franzen, Martin et al. essentially gave an interpretation of the previous mechanistic hypothesis proposed by Petrich, Martin et al. [*Biochem.*, 27, 4049–4060 (1988)], where in Eq. 2 they proposed again a triplet state $HbCO(^3T_1)$ populated photochemically in <50 fs prior to the dissociation of $CO(^1\Sigma)$:

Reprinted adapted with permission from *Biochemistry*, 1988, 27 (11), pp 4049–4060. Copyright 1988 American Chemical Society.

In our opinion, this model was the result of excluding vibronic coupling and spin-orbit coupling as the main mechanism of dissociation, as Franzen et al. concluded by symmetry arguments that vibronic couplings should be negligible. They rather invoked a valence tautomerism (VT in the passage quoted above) as the plausible explanation. Our simulations show that vibronic couplings are strong enough to lead to an ultrafast dissociation, that takes place in ~ 20 fs. Strong vibronic coupling at the origin of ultrafast processes has been reported nowadays in many photochemical processes ranging from small molecules (like photoisomerizations) to materials (like singlet fission).

We have modified our introduction to include the citation of Petrich and co-workers and the transient absorption spectra as the referee suggested. However, we maintain our sentence on the disagreement

between Franzen et al. and Head-Gordon et al. If the referee can provide us with a specific citation of the experimental works claiming singlet photolysis, we are willing to cite it in our manuscript, but we are not aware of such a reference. In the introduction, we cite textually the sentence from the Franzen et al. paper to avoid confusion at this point. The second paragraph of the introduction is modified as follows (modifications in blue):

The heme-CO photolysis is an ultrafast process. Recent pump-probe X-ray experiments of myoglobin, with initial pump pulse exciting the 1Q state, seem to agree on a two-step kinetic reaction: (i) a first step taking <50-70 fs, attributed to both CO photolysis and partial SCO, and (ii) a completion of the the spin transition to the HS state in ~300-400 fs.^{5,6,20} Despite the numerous studies on heme-CO photolysis, the kinetics and mechanism of dissociation are still under debate, notably regarding the ultrafast nature of the reaction, and the spin and character of the photolytic state. The rate of photolysis has never been experimentally reported, apart from the upper bound of 50-70 fs. As far as the photolytic state is concerned, the most widely accepted hypothesis is that dissociation occurs from a metal-ligand charge-transfer (MLCT) state.^{15-18,20} Still, experiments and theory do not provide a unified picture to date. On the one hand, Franzen, Martin et al., based on time-resolved absorption and Raman experiments, described a “rapid spin state change that must precede photolysis”.²⁰ They consider that photolysis is occurring from a triplet metal→porphyrin ring transfer (3MLCT dissociation).^{20,21} In the model by Franzen et al., the ultrafast reaction is due to valence tautomerism. This mechanism implies a rapid interconversion of several quasi-degenerate electronic states involving d-transitions, some of which are dissociative for the CO bond. On the other hand, Head-Gordon et al., employing time-dependent density-functional theory simulations, proposed a dissociation in a singlet metal→ σ_{CO}^* state (1MLCT dissociation).¹⁵⁻¹⁸ In the model of Head-Gordon et al., CO photolysis occurs in a Marcus-like process, in which the initial 1Q population is transferred to the photolytic state after crossing a barrier of <0.2 eV. However, this hypothesis seems in contradiction with experimental observations of absence of fluorescence emission, an ultrafast reaction and a unity quantum yield for the heme-CO complex.²² The theoretical method used provided an incomplete description of ultrafast heme-CO photolysis, since the coupled electron-nuclear motion was neglected. Recent theoretical studies of similar iron complexes indicate the fundamental role of nuclear motions in explaining the ultrafast nature of intersystem crossing (ISC).²³⁻²⁵ Such electron-nuclear strong coupling has been experimentally observed in time-resolved X-ray absorption in a Fe(II) tris(2,2'-bipyridine) complex^{26,27} and also in ferricyanide ion.²⁸

The authors state that their study elucidates the importance of vibrational modes for photolysis. However, Figure 3 shows three modes (1) a Fe-CO stretch (2) a Fe-C-O bend and (3) a Jahn-Teller distortion. The authors state that the Jahn-Teller distortion is not important for photolysis, which is something that is obvious from the symmetry of the problem.

Response: We agree that perhaps this part was not clear enough in our previous manuscript version and we have improved the wording. As the referee says, it is clear from the symmetry of the problem that Jahn-Teller symmetry breaking is not able to couple directly to the dissociative mode. However, symmetry breaking is fundamental to rapidly transfer the population from 1Q state population to the photolytic MLCT state. Indeed, we show that this transfer has a kinetic rate of 26 fs on average. Head-Gordon et al. predicted a barrier of 0.2 eV for such transfer, which would not allow for an ultrafast reaction.

We included the following sentence referring to symmetry-breaking Jahn-Teller modes:

Such a vibration is not directly coupled to the Fe-CO bond, but is fundamental to induce the interstate coupling between the Q-band and the MLCT band. The relaxation energy induced by such vibrations can be as large as 0.5 eV, which rapidly leads the wavepacket far from the Franck-Condon region.

One does not need a calculation at the CASSCF(10,9)/CASPT2/ANO-RCC-VDZP level in order to establish which mode is most important here.

Response: It is correct that some qualitative insight can be obtained regarding the dominant modes using symmetry considerations and more approximate methods. However, a high-level electronic structure

treatment is mandatory to capture the interactions between many vibronically coupled states of different multiplicity. The use of CASPT2 is fundamental to obtain accurate singlet, triplet and quintet bands. This is one of the main strengths of our methodology, since we show that multiconfigurational effects in the excited states are very important. These were completely neglected in any of the previous theoretical studies.

To justify the use of CASPT2, we have included the following sentence in the revised manuscript, referring to the ^{1,3,5}MLCT bands:

These bands of states have strong multiconfigurational character, and were not correctly represented in previous single-reference studies based on density-functional theory.^{15,18}

The point it is not clear what new information is being presented in this manuscript. Indeed, the wave packet study is a nice contribution that reminds us of an interesting problem, but the justification for publication in Nature is lacking.

Response: We hope that this letter and the revised manuscript will convince the referee of the novelty of our model to determine the complex mechanism of heme-CO photolysis. To summarize the new information we provide: we have performed the most advanced simulation of heme-CO to date. Our model reproduces experimental kinetic rates, we reported features of the photolysis never discussed before, and we give the most detailed interpretation of the vibronic mechanism at the origin of the ultrafast photolysis. We established a protocol of simulations that will become the state-of-the-art for organometallic complexes and other systems in which vibrational motion plays a fundamental role.

The theoretical protocol and the conclusions of the heme-CO photolysis can interest a broad audience of chemists, biologists, and physicists, and therefore we believe it matches the interests of the audience of Nature Communications.

Reviewer #3 (Remarks to the Author):

I have read the manuscript as well as the previous reports and rebuttals. To make it short:

I wish I would have done this work!

Why? It is not only time (there are many papers appearing on spin-crossover these days), but it also must be considered as providing a theoretical benchmark to the drosophila system for studying photo triggered protein dynamics.

Apparently, the response to reviewer #1 did improve the presentation substantially. W.r.t. reviewer #2, I obviously agree that this is excellent work, but I cannot share his/her view that the readership will come from a specialized audience only.

To wrap it up I recommend publication, provide that the authors have addressed the following points:

- literature:

There is a large body of work by Chris Meier on the quantum dynamics of this molecule & I wonder why this has been ignored. Further, concerning spin-crossover and the role of vibrations it might be worth citing the recent photoelectron study on Fe(CN)6 by Aziz and coworkers in PCCP 19, 14248 (2017)

Response: The literature on heme-CO is very rich, directly proportional to the complexity of the physical processes occurring in this system. Since it is impossible to review and cite the whole literature, our initial idea was to restrict the discussion specifically to the ultrafast photolysis and the excited electronic states.

We agree with the referee, though, that the work of Christoph Meier deserves citation, as he applied for the first time quantum wavepacket dynamics to carboxyhemoglobin for designing IR pulses using quantum control. We included the citation of J. Chem. Phys. 123, 044504 (2005) and J. Phys. Chem. A, 117, 12884 (2013) in the introduction (see Refs. 11 and 12 in the new manuscript).

We also thank the reviewer for the suggestion of Aziz and coworkers work. We were unaware of this recent publication, and we are happy to find similar iron complexes with ultrafast Jahn-Teller effect like what we find for heme-CO. We have cited this work in the introduction as another example of strong electron-nuclear coupling to explain the photochemistry of iron complexes. We introduced the following sentence in the manuscript (new modifications in blue):

Such strong electron-nuclear coupling has been observed in time-resolved X-ray absorption in a Fe(II) tris(2,2'-bipyridine) complex^{26,27} and also in ferricyanide ion.²⁸

- technical:

I did not fully understand the model Hamiltonian. First, Eqs 4-5 suggest that anharmonic mode coupling has been taken into account in a rather unusual way, i.e. thinking of a Taylor expansion of the exponential. At the same time I doubt that this is actually what has been done since only diagonal cuts of the adiabatic potentials are represented. If mode coupling has been neglected, please motivate!

Response: We have decided to eliminate any reference to the word anharmonicity from our text, as it apparently induces to ambiguities. We did not refer to anharmonic coupling between vibrational modes, but rather to anharmonicities of the potential energy surfaces of excited states along each of the ground-state normal modes. Our present model employs the vibrational modes of the ground state, and therefore not only anharmonic vibrational couplings have been excluded, but also Duschinsky rotation effects, which accounts for the mixing of vibrational modes in the excited state.

We excluded anharmonicities and Duschinsky effects because these cannot be easily computed for 180 excited states, especially when they form quasi-continuous bands. Indeed, both the Duschinsky rotation matrices and anharmonic couplings require to first obtain harmonic frequencies of excited states, which in turn require well-defined minima in the electronic excited states. From our experience, it is quite difficult to construct such a model with so many excited states including these effects.

At this point, we cannot give an estimate of the effect of Duschinsky rotations, but we can compute the anharmonic couplings in the ground state (see Supporting Material). We observe that such anharmonic couplings are smaller than the non-adiabatic and spin-orbit couplings. Indeed, we obtained that on average the anharmonic couplings were smaller than 15 cm^{-1} , which are much smaller than the non-adiabatic couplings and spin-orbit couplings, which can be as large as 500 cm^{-1} . In addition, given the ultrafast nature of this reaction ($< 50\text{-}70 \text{ fs}$) we can assume that vibrational coupling effects are not yet important on this time scale. Therefore, we consider that the electronic couplings dominate and restricted our model to 1D cuts.

We would like to point out that, even though our model does not systematically include anharmonic effects, the frequencies we use for the model do not come from the harmonic approximation, but result from a parameterization that realistically reproduces the potential energy surface cuts for each excited state and normal mode.

We have included the following remark in the manuscript to clarify the main approximations of our model:

Duschinsky rotation effects and anharmonic couplings between vibrational modes have been neglected in our model; in particular, the latter are negligibly small compared to the vibronic couplings (see further Supporting Information).

We have also included a Table of the anharmonic couplings in the Supporting Information (see Sec. III.C in page 16). As it can be seen, the dissociative mode (mode 33) has strong anharmonic coupling with the doming mode. However, it is not anharmonically coupled to the Jahn-Teller modes. This clarifies further our remark in the text that the doming mode is activated after photolysis.

How the neglected coordinates in the 1D cuts have been treated, i.e. frozen or relaxed? In any case, motivate the choice.

Response: The coordinates have been frozen. For an ultrafast photochemical process, this is in general the natural choice because many of the low-frequency, weakly coupled modes remain close to the initial ground-state geometry. We selected the normal modes in our model in two ways: (i) by expansion in normal modes of the difference of the High-Spin minimum and the Low-Spin minimum geometries, as these are the two limiting structures before and after the photoreaction, and (ii) by identifying the modes inducing the largest reorganization energy in the excited state. The resulting set of modes gave the most stable propagation, with small errors in the numerical integration. We have included the following sentence in the text:

The latter are obtained by one-dimensional potential energy surface cuts along each ground-state normal mode direction. The passive normal modes have been kept frozen along the cuts.

From the same equations I would gather that the dissociative coordinate is taken to correspond to

the Fe-C(O) stretching normal mode. Practically, normal mode displacements are only meaningful in the vicinity of equilibrium structures. that means even small displacements of other atoms which are not part of the actual Fe-C(O) stretching, but exist because the Fe-C(O) motion is not fully decoupled, might cause some artificial behavior if the system is stretched too far. Is this perhaps the reason for the short time approximation or the limited range of distances in Fig. 2?

Response: This is true. For the first kinetic step (the main scope of this study), we are confident that vibrational mode couplings do not play a role, because the photoreaction is too fast. This is rather controlled by non-adiabatic, spin-orbit couplings and the large reorganization energies of certain vibrations on the excited state potentials. For the second kinetic step, mode couplings certainly start to become more important. In this range, as demonstrated in previous work involving two of us (K. Falahati and I. Burghardt) [Science, 350, 445 (2015)], also coupling to the protein modes becomes important. To have a more realistic model for longer time, it would be necessary to include both protein modes and anharmonic couplings in the excited states, but currently this is beyond our present capabilities. Our model is correctly capturing the spin-crossover mechanism (the vibrational modes have been selected to describe the whole photoreaction), and that is why the second rate compares very well with experiments. However, the geometric features, are only approximately described since a number of modes that become important at longer times have not been included.

- presentation:

Fig. 1: I don't think two digit numbers are justified in panel b.

Response: Indeed, we have modified the figure with rates that have been rounded with no decimals.

Fig 1. caption: "Period of oscillation is initially 0.9 A", What's meant here?

Response: This was a mistake. We refer to the amplitude of the motion. We have corrected it in this new manuscript version.

Fig. 1: the shaded area was not visible upon printing

Response: We could reproduce this problem when we changed the operating system. We realized that shaded areas were not correctly exported in the pdf, so we exported first in png and embedded them in the pdf. Now the appearance should be correct.

line 140: Did the initial geometries break the symmetry? How about temperature? Is this supposed to mimic finite T as well?

Response: Yes, the symmetry is broken in the initial geometries. For parameterizing the model Hamiltonian, we took as reference the minimum energy structure of the ground state, from which we use the normal modes to expand the model Hamiltonian. Thus, the initial symmetric structure is recovered in the model by setting all vibrational coordinates to zero $\{Q_i=0\}$. In the first draft of this manuscript, we performed a single quantum dynamics simulation starting from the fully symmetric structure only.

The remarks of the referees pushed us to design a strategy to include in the initial conditions of our quantum dynamics the effects of symmetry breaking and inhomogeneous broadening due to the fluctuations of the protein. What therefore took several MD snapshots, for which we extracted the atoms corresponding to our molecular model. Then, we projected the difference of the MD geometry with the reference geometry of our model and expanded this in normal modes. This gave us 10 sets of initial conditions $\{Q_i\}$ that reproduced the MD geometries. We took this to start 10 quantum dynamics simulations and we performed a statistic of the evolution of populations and geometrical parameters. The details of this procedure are described in the Supporting Information, page 16.

This protocol is the best we can do for now. We thus include the effects of symmetry breaking of the initial geometry, but only indirectly the effect of temperature due to the fluctuations of the protein. In the future, we plan also to include the vibrations of the protein in the model Hamiltonian to be able to reproduce the temperature effects more precisely, but this is currently far beyond our possibilities and less relevant for ultrafast photochemistry.

We have clarified this part in the new version of our manuscript:

The results have been obtained by sampling 10 independent quantum dynamics simulations with different initial conditions. The initial conditions are obtained by projecting the heme-CO geometries extracted from molecular dynamics snapshots of a myoglobin protein onto our vibronic model Hamiltonian (see Sec. IV of the Supporting Information for further details.)

line 149 and further on: The unit of a rate is 1/time, better call the numbers time constants or alike

Response: We have modified this in the article and refer to time constant instead of kinetic rate.

line 171: Fe-C(O) stretching frequencies are apparently state dependent. What do we learn from this statement which is emphasized in several places?

Response: The harmonic frequency of oscillation of Fe-C(O) in the ground state is 488 cm^{-1} , thus leading to a period of oscillation of 68 fs. However, when the wavepacket propagates in the excited states, this bond oscillates at $\sim 800\text{ cm}^{-1}$, thus leading to a period of 42 fs. What we want to stress is that the effective frequency of oscillation of certain vibrations in the excited states can be larger than in the ground state, due to the vibronic couplings. This is also indicating that this reaction is faster in the excited state than in the ground state.

line 401: "entropic trapping". I have difficulties to accept this characterization given the fact that a reduced dimensional model is considered at a very short time scale, i.e. there is little chance that possible micro states have been fully explored yet.

Response: We agree that employing such a concept is not rigorously justified on the grounds of our calculations. Therefore, we have eliminated any reference to entropy from our manuscript.

REVIEWERS' COMMENTS:

Reviewer #3 (Remarks to the Author):

I am happy with the response of the authors and recommend publication.